# Real-time concrete strength monitoring using piezoelectric sensors and deep learning

Guangshuai Han [1], Yen-Fang Su [1,2], Rui He [1], Cihang Huang[1], Zhihao Kong[1], Guang Lin [3,4], Yining Feng [1] ✉ & Na Lu [1] ✉

This study presents a transformative advancement in civil engineering by integrating artificial intelligence with infrastructure sensing to redefine concrete structures testing and monitoring. Traditional methods for evaluating concrete performance, largely unchanged for over a century, rely on labor-intensive, proxy-based techniques that are both time-consuming and limited in reliability. Our approach combines using piezoelectric sensors with AI-driven data analysis to enable real-time, in situ monitoring of structural conditions with enhanced accuracy and automation. By employing deep learning models to interpret electromechanical impedance signals, the system eliminates the need for destructive testing or human intervention, offering a scalable solution suitable for real-world deployment. Successfully validated across four large-scale highway construction projects, the system demonstrates prediction errors within approximately 15% when benchmarked against standard compression tests conforming to ASTM C39. Aspects of this technology, such as the underlying sensing principle have been incorporated into a new standard by the American Association of State Highway and Transportation Officials (AASHTO T412), representing a significant step toward the national standardization of this non-destructive testing method. Our findings propose a scalable method to integrate intelligent sensing into civil infrastructure system. This will enable the development of resilient and sustainable infrastructure, moving beyond traditional infrastructure monitoring.

Concrete is the backbone of civil infrastructure, essential to public safety, transportation, and industrial development[1–4]. Yet, the primary method for assessing its most critical property—compressive strength—has remained unchanged since 19th century. Over the past century, numerous efforts have been made to replace or enhance this antiquated approach, aiming to improve efficiency, accuracy, and provide a true assessment of the concrete structure itself, rather than relying on proxy measurements or destructive testing[5,6]. Non-destructive testing (NDT) techniques such as the rebound hammer test (ASTM C805), penetration resistance method (ASTM C803), pull-out test (ASTM C900), ultrasonic pulse velocity (UPV) (ASTM C597), and maturity test (ASTM C1074) have all been explored[7–12]. However, most NDT techniques demand specialized expertise, are often labor-intensive, and require extensive calibration, driving up project costs and extending timelines[13]. Consequently, the most-commonly used proxy method remains entrenched in engineering practice, as it offers a conservative and inefficient approach[14]. This reliance comes at the expense of time and resources, forcing

[1]Lyles School of Civil and Construction Engineering, Purdue University, West Lafayette, IN, USA. [2]Department of Civil and Environmental Engineering, Louisiana State University, Baton Rouge, LA, USA. [3]Department of Mathematics, Purdue University, West Lafayette, IN, USA. [4]School of Mechanical Engineering, Purdue University, West Lafayette, IN, USA. ✉e-mail: feng109@purdue.edu; luna@purdue.edu

engineers to prioritize safety by defaulting to a method that is nearly a century old.

C. Liang was a pioneer in demonstrating that piezoelectric materials, when bonded to a structure, can harness electromechanical coupling to effectively capture and analyze the mechanical behavior of a structure[15]. By utilizing electromechanical impedance (EMI) signals generated by piezoelectric sensors, the mechanical characteristics of structures can be digitized and continuously monitored in real time[16,17]. Due to their high sensitivity and capability for in-situ monitoring, the EMI-based approach using piezoelectric sensors has seen widespread application in the health and strength monitoring of concrete structures[18,19]. However, the full application of this technology in concrete remains at the stage of feasibility validation. Recent investigations have primarily focused on establishing that EMI signals are strongly correlated with changes in the structural condition, rather than directly providing precise measurements results of concrete strength. A persistent challenge lies in the inherent variability of piezoelectric sensors, which impacts the accuracy and reliability of the measurements[20]. Furthermore, the complexity of concrete—characterized by varying cement compositions, aggregates, water-to-cement ratios, and admixtures—creates a multifaceted strength development process that is difficult to monitor consistently[21–23]. Additionally, the early-stage hydration heat in large-scale concrete structures introduces significant temperature fluctuations, which in turn affect the EMI signals captured by piezoelectric sensors[24–26]. These effects, combined with practical challenges in sensor deployment during construction, have impeded the practical adoption of piezoelectric sensors for real-time strength monitoring in concrete.

With advancements in data science, techniques such as machine learning, deep learning, and artificial intelligence (AI) have increasingly been utilized to process complex datasets and have been explored in EMI-based approaches for piezoelectric sensor monitoring[27–30]. These approaches have been applied not only for compressive strength estimation, but also for damage detection[31,32], damage classification[33], and corrosion assessment[34] using EMI signals, with some studies further integrating AI algorithms to enhance sensitivity for understanding micro-level structural deterioration. However, most existing studies have been exclusively conducted in controlled laboratory settings, where small-scale samples eliminate the effects of temperature fluctuations and ensure uniform curing conditions. As a result, the generated datasets often lack the real-world variables in construction projects, such as sensor-to-sensor discrepancies, ambient temperature fluctuations, and changes in material properties. Moreover, these limited lab scale database approaches are typically constrained by limited sensor setups and a narrow selection of concrete types, further restricting the generalizability of the models to actual engineering conditions[18].

In addition to leveraging AI for processing EMI signals, research efforts have also attempted to use AI to predict concrete performance based on mix design[35,36]. Yet, the existing mix-design-based method has inherent restrictions. In principle, these approaches are built on the assumption that a given mix design will always result in similar chemical composition, hydration behavior, and, ultimately, strength development. However, this assumption does not account for real-world complexities, where binder type, aggregate source, water quality, and environmental exposure can significantly alter hydration kinetics—even when the nominal mix design remains the same. For instance, supplementary cementitious materials (SCMs) and admixtures used in concrete have countless variations, each with distinct chemical compositions and varying impacts on concrete performance[23,37]. Additionally, the chemical composition of cement and the properties of fine and coarse aggregates vary greatly by region, making it difficult to generalize models suitable for different geographies[38]. As a result, the development of universally applicable databases and predictive models built on chemical composition inputs remains highly challenging and often ineffective for field conditions. In contrast to mix-design-based approaches, piezoelectric sensing combined with electromechanical coupling enables a direct mechanical-to-mechanical mapping. In principle, the EMI signal captures changes arising solely from variations in the mechanical properties of the host structure, without relying on assumptions about chemical composition or hydration behavior. Therefore, the extracted information is inherently unaffected by variations in chemical composition, curing conditions, or other human factors, making it robust, broadly applicable, and well-suited for diverse field conditions.

The present study focuses on our efforts to establish a comprehensive, real-world engineering database for piezoelectric-based concrete strength monitoring, as shown in Fig. 1a. Over 100 piezoelectric sensors were strategically placed in the slab and on cylindrical samples to monitor strength development over a one-year period. Figure 1b illustrates the distribution of the sensors and experimental concrete blocks, providing a detailed overview of the deployment for comprehensive data collection. As depicted in Fig. 1c, we introduced a baseline mechanism to compensate for the intrinsic errors of the piezoelectric sensors, ensuring more reliable signal interpretation. Additionally, we incorporated non-signal features into the 1D Convolutional Neural Networks (1DCNN) architecture to account for variations in the concrete curing conditions, addressing the complexities arising from environmental and material factors. These resulted in highly accurate strength predictions. Furthermore, we performed extensive model analyses to clarify how the deep learning algorithm processes sensor signals, integrating these compensations to predict concrete strength with high precision. In four independent field tests, involving more than 30 sensors, the model-trained solely on laboratory slab data successfully predicted concrete strength under actual real-world conditions with sufficient accuracy. Specifically, the predicted values from the sensors deviated from the proxy sample measurements by 1 to 2.5 MPa, corresponding to a variation of 10% to 25%. This falls within the normal range of errors for practical engineering applications.

Overall, this study presents an end-to-end process—from database construction and model development to field validation—for enabling real-time, in-situ concrete strength monitoring using piezoelectric sensors and deep learning. Through extensive datasets and AI modeling, we have verified the hypothesized relationship between the first peak in EMI signals and concrete strength, providing a clear explanation of how the sensor feedback correlates with structural performance. Additionally, we demonstrate a successful large-scale field trial of AI integrated NDT, moving beyond laboratory experiments to real-world construction environments. This work has progressed over a span of seven years, from initial theoretical research to the completion of technical transfer and real-world application (see technology roadmap in Supplementary Fig. S1). While the core sensing principle has informed the establishment of the American Association of State Highway and Transportation Officials (AASHTO, T 412-24) standard, the piezoelectric-based concrete strength sensing method presented here is currently undergoing field trials in over 34 U.S. states, offering a more efficient solution for civil infrastructure projects. This development strategy also provides a generalizable framework for solving similar engineering challenges and lays the foundation for the intelligentization of infrastructure systems through real-time, AI-assisted structural monitoring and enhanced construction reliability.

## Results
### Full-Scale Experimentation and Data Infrastructure
We cast seven large concrete slabs, each measuring 8 feet by 12 feet, in an outdoor environment to collect data on concrete strength at various ages, as well as EMI signals captured by piezoelectric sensors. The data collection spanned over a period of one year, with the goal of establishing an informative, diverse, and comprehensive database for strength development in concrete structures. As shown in Fig. 2a, the

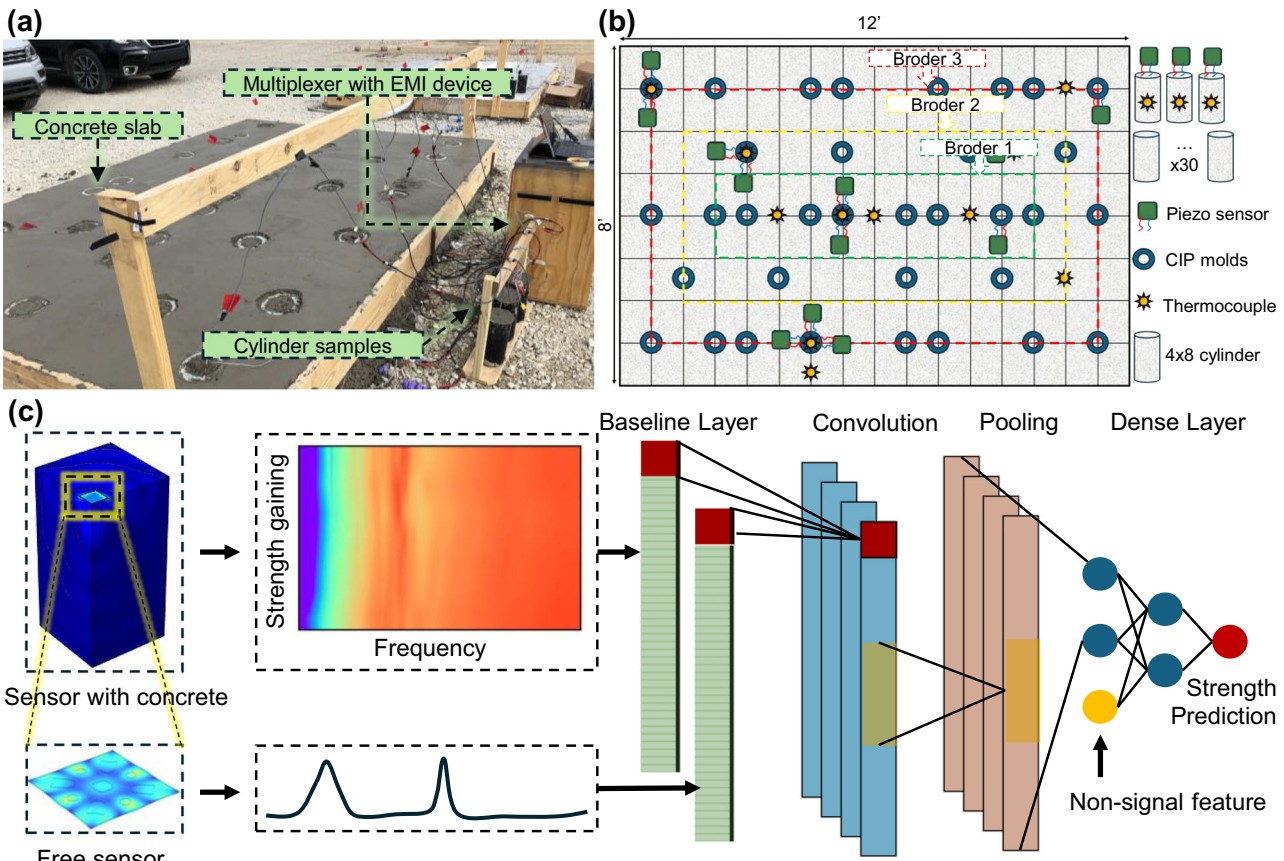

**Fig. 1 | Real-world concrete monitoring diagram with the proposed AI framework for infrastructure sensing. a** Concrete slab cast with piezoelectric sensors and cylinder samples, designed to capture in-situ conditions for long-term monitoring. **b** Sensor and sample layout showing the placement of over 100 sensors across the concrete slab and associated test cylinders (including cast-in-place (CIP) molds and cylinder samples). **c** Flowchart of the AI-based signal processing model, incorporating baseline compensation and non-signal features for accurate strength prediction.

violin plots display the strength data collected from each slab. The specific mix designs for each slab, including water-to-cement ratios, admixtures, and SCMs, are detailed in Table S1. These mix designs were based on standard pavement concrete mixtures commonly used by the Indiana Department of Transportation (INDOT), with a target 28-day-compressive strength of 4000 psi (approximately 27.58 MPa). Slabs 5 exhibited compressive strengths closest to the target, as their compositions most closely followed typical INDOT specifications. Slabs 1–3 showed significantly higher 28-day strengths due to the incorporation of nano-silica, a material previously shown in our studies to enhance strength development[39]. Slab 7, on the other hand, was deliberately designed with a higher water content to simulate underperforming or failed concrete conditions in practical construction scenarios.

Within this practical design framework, we made controlled adjustments to the proportions of key materials to generate slabs with distinct strength development profiles. This strategy enabled us to test the generalizability of the proposed method across different concrete with various mixtures. Environmental conditions for each casting date, including ambient temperature and wind speed, are shown in Fig. S2 of the Supplementary information. Each slab was cast with a unique mix design on one of three separate dates, under varying environmental conditions. By incorporating realistic variation in both curing conditions and mix composition, we aimed to closely replicate field casting scenarios. This approach allowed us to build a representative and variable-rich dataset that reflects the diverse factors influencing concrete strength in real-world construction environments.

Within the same concrete slab, we employed two experimental configurations to create distinct strength profiles. Compressive strength measurements were obtained from two types of samples: cast-in-place (CIP) cylinders following ASTM C873 and the more commonly used 4-inch by 8-inch (4 × 8) cylinders cast outside the slab, as per ASTM C39, which served as proxy samples for estimating the concrete strength[40,41]. Correspondingly, we deployed piezoelectric sensors in two matching configurations: one set embedded directly into the concrete slab to correlate sensor signals with the CIP sample strength and temperature, and another set embedded into the 4 × 8 cylinders to map sensor data to the proxy sample conditions. This design ensured that the sensor signals reflected the curing conditions and material composition associated with each type of strength label.

The primary motivation behind this experimental design was to address a well-documented issue in current engineering practice: proxy samples often fail to accurately represent the true strength of large-scale concrete structures. This discrepancy arises because mass concrete, such as slabs or pavement sections, undergoes extensive hydration and retains significant heat, which accelerates strength development. In contrast, smaller proxy samples like 4 × 8 cylinders do not exhibit the same thermal behavior, resulting in a mismatch between their measured strength and the actual in-situ concrete performance. This discrepancy was observed in our experiment as well. In Supplementary Fig. S3, we present the relationship between temperature and ambient conditions over a one-year period for both sample types. It is evident that after a few hours of hydration, the temperature in CIP samples far exceeds that of the 4 × 8 cylinder

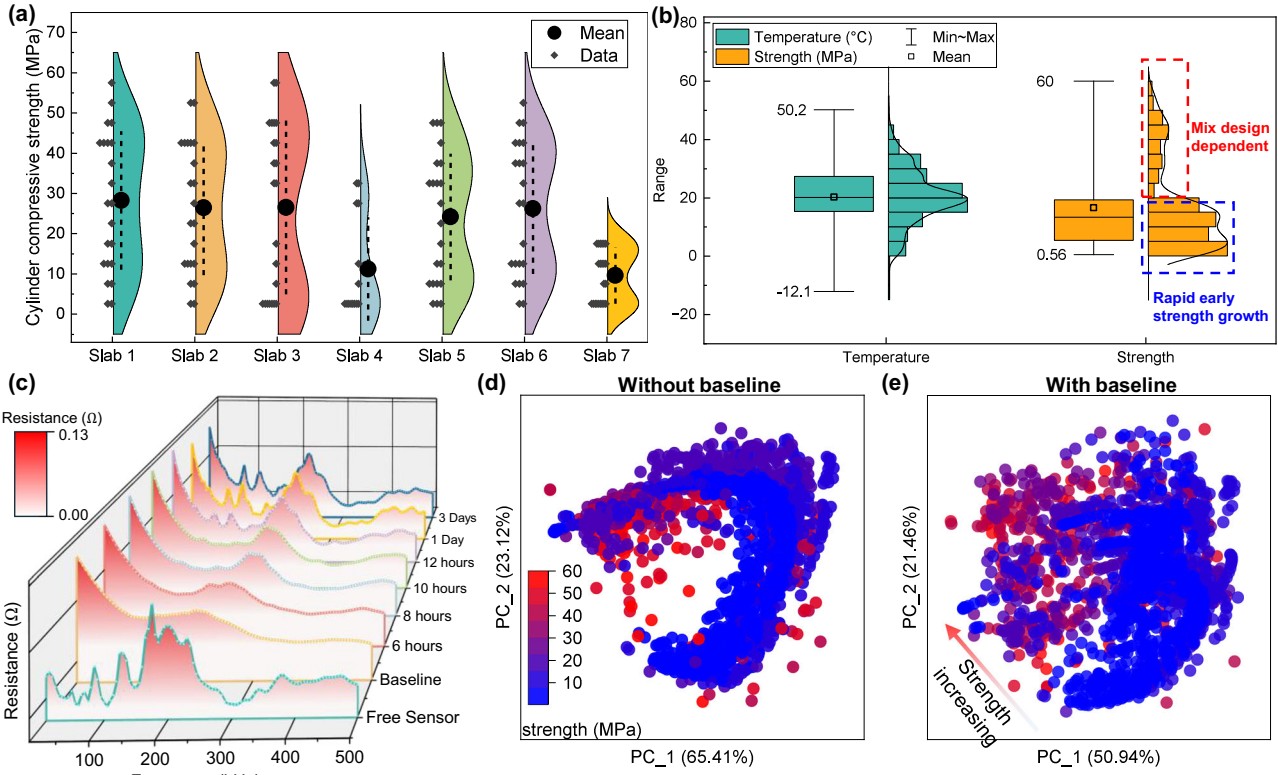

**Fig. 2 | Strength sensing data collection and visualization for concrete structures. a** Strength distribution across seven concrete slabs at various curing ages, representing different strength development patterns captured in the dataset. **b** Histogram of the overall dataset, featuring both internal temperature and compressive strength measurements. The temperature follows a normal distribution, influenced by external environmental factors and concrete's exothermic hydration process. The strength data can be segmented into two categories: early-age concrete, which was densely sampled, and later-age concrete exhibiting a normal distribution, shaped by different mix designs and curing conditions. **c** EMI sensing performance from a single representative sensor, showing signal variations as the concrete ages, with clear changes over time. **d** Principal component (Principal component) analysis of real-time concrete sensing EMI signals only. Here, PC_1 captures the dominant trend of strength-related signal evolution, while PC_2 represents secondary variations in the EMI response. **e** Principal component analysis of both real-time EMI signals and baseline EMI signals, highlighting the high correlation between EMI signal patterns and concrete strength development.

samples. Due to their smaller volume, the proxy samples are more easily influenced by ambient conditions, with their internal temperatures quickly approaching the ambient temperature. The difference in temperature between mass concrete and proxy cylinder samples affects the hydration process, leading to variations in the measured strength. This phenomenon has also been reported by the Alabama Department of Transportation (DOT), and we compared our findings with theirs in Supplementary Fig. S4a[42]. Their data shows that CIP molds exhibit higher strength than the 4 × 8 cylinders. Additionally, since the 4 × 8 cylinders were cured in insulated chambers, their conditions differ from those of the concrete slabs, resulting in no clear linear relationship between the two. These findings underscore the unreliability of using proxy cylinder samples, highlighting the need for an in-situ testing method that can accurately measure the true strength of concrete in practical applications. Moreover, their results show that the strength of CIP samples located on the edge of slabs closely mirrors the strength of CIP samples located in the center of slabs, suggesting that the strength distribution within the entire slab is relatively uniform. Our findings further demonstrate that CIP molds consistently exhibit higher strength than the 4 × 8 cylinder samples as shown in Supplementary Fig. S4b. Unlike in previous studies from Alabama DOT, we did not use insulated chambers for the 4 × 8 cylinders; instead, we placed them adjacent to the concrete slab, exposing them to the same environmental conditions. This discrepancy arises because the CIP molds are embedded within the mass concrete, where the retained heat promotes hydration. In contrast, in the Alabama DOT study, the use of insulated chambers for the 4 × 8 cylinders allowed for greater retention of hydration heat, preventing rapid dissipation to the surrounding environment. Consequently, it can be concluded that the commonly applied proxy sample method in engineering practice does not accurately reflect the mechanical properties of the concrete slab. This underscores the urgent need for in-situ NDT methods to better capture the true behavior of concrete in real-world applications.

Figure 2b illustrates the distribution of internal sensor temperature data and the concrete strength labels within the dataset. The temperature data displays a normal distribution, with the median and mean values centered around 20 °C. The overall temperature range spans from a low of −12.1 °C to a high of 50 °C, capturing the extreme conditions that may occur in real-world engineering scenarios, such as winter construction or elevated temperatures caused by hydration in mass concrete structures[25,26,43]. Since the EMI signals in concrete strength sensing are influenced by both the mechanical properties of the structure and temperature variations[26], these temperature data points are critical for avoiding errors in strength prediction and must be considered in the model. The distribution of concrete strength data can be separated into two distinct sections. Below 20 MPa, the strength measurements are densely distributed, a result of the experimental design, which focused on intensive data collection during the first 24 h after casting. During this period, concrete undergoes significant strength gain, and the corresponding EMI signal changes rapidly[44]. This intensive sampling during the early stages is crucial for capturing the rapid evolution in EMI signals and establishing a robust mapping between EMI signals and strength, which aids in the learning process of the deep learning model. For strength values exceeding

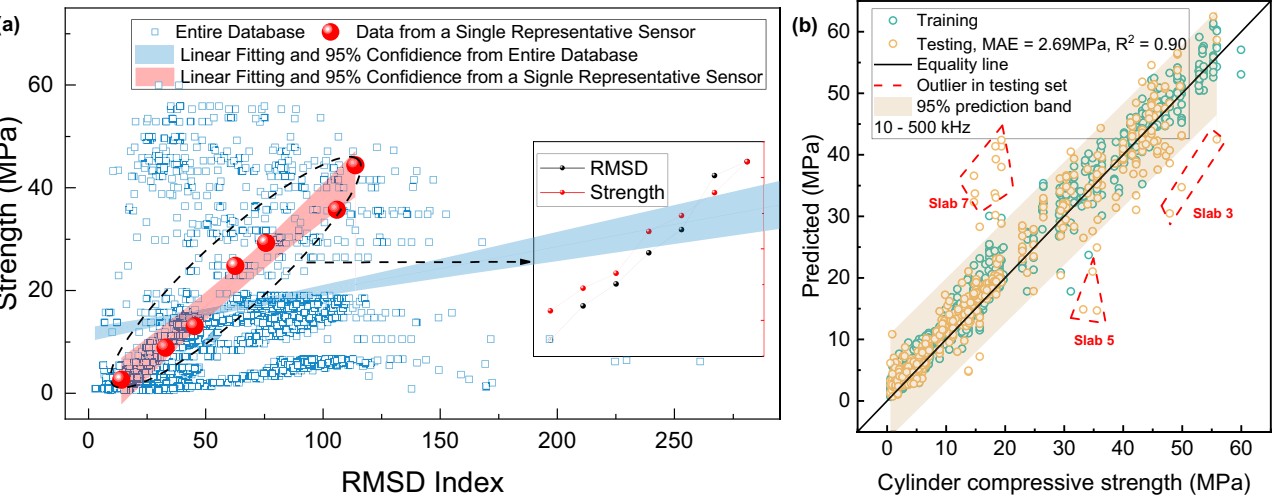

**Fig. 3 | RMSD–strength relationship and CNN-based strength prediction performance.** (**a**) Relationship between the root mean square deviation (RMSD) index and concrete strength using data from the entire EMI signal dataset. The blue squares represent data from the entire database, and the red circles represent data from a single representative sensor; (**b**) Comparison between predicted strength and actual strength using the proposed baseline mechanism combined with a 1D CNN model, evaluation matrix including correlation coefficient ($R^2$) and mean absolute error (MAE).

20 MPa, the data follow a normal distribution, reflecting the use of various mix designs and curing conditions.

Figure 2c presents the sensing results from a representative piezoelectric sensor. Before the sensor was deployed in the concrete structure, two main resonance peak clusters are observed in the 5–500 kHz frequency range. Once the sensor was embedded in the concrete structure, we began signal collection at the 4-h mark of the concrete curing process, using this time point as the baseline for strength prediction. Prior research has demonstrated that signal collection at 4 h can effectively track concrete strength development patterns[13]. In the baseline signal, the peaks diminish significantly, displaying very low amplitudes. This is because, at early ages, concrete behaves more like a viscoelastic material, absorbing most of the mechanical waves emitted by the piezoelectric sensor[45,46]. As the concrete continues to cure, exceeding the 10-h mark, the signal peaks start to change, evolving from a single peak to twin-peak or multi-peak patterns. This transition indicates that the concrete is transforming from a viscoelastic material to an elastic one, with reduced wave energy absorption[44,45]. These results demonstrate that the piezoelectric sensor's EMI signals can effectively capture changes in the mechanical properties of the concrete. This correlation arises from the electromechanical coupling principle underlying EMI sensing: as the mechanical impedance of the host material (i.e., the concrete) evolves during curing, it alters the boundary conditions experienced by the piezoelectric patch. These changes in mechanical impedance directly affect the electrical admittance spectrum of the sensor, especially the amplitude and position of its resonance peaks. Thus, variations in concrete stiffness and damping characteristics are encoded in the EMI signal, enabling real-time monitoring of mechanical property development. Additional EMI signals collected over time from the seven concrete slabs and different sensors, showing how the signals evolve with age, can be found in Supplementary Figs. S5–S18.

To move beyond representative sensor examples and more effectively illustrate the overall relationship between EMI signals and concrete strength, we applied Principal Component Analysis (PCA) to reduce the dimensionality of the full EMI dataset. Figure 2d shows the result using only real-time EMI signals, while Fig. 2e shows the PCA projection after combining real-time signals with baseline signals. In Fig. 2d, the data distribution appears more scattered, with irregular clustering and a less coherent trend along the principal components. By contrast, Fig. 2e demonstrates a clearer strength gradient along the

PC_1 axis and exhibits a more compact structure. The reduced number of isolated outliers in Fig. 2e suggests that introducing baseline signals effectively mitigates variation caused by sensor-to-sensor differences, resulting in dataset that are more consistent and more amenable to machine learning models. Nevertheless, the projection of EMI signals onto the principal component space does not exhibit a fully coherent trend with strength. Such a result is anticipated, as simple linear dimensionality reduction cannot fully capture the complex nonlinear relationship between EMI signals and strength. In addition, the current analysis does not account for temperature, which is known to influence EMI signals and likely contributes to the remaining dispersion and outliers in the projection. Overall, we found that incorporating baseline signals improves the structure and consistency of the dataset. However, advanced signal processing methods are still required to effectively extract the relationship between EMI signals and concrete strength, as will be elaborated in the following section.

## Model development and optimization

Before applying deep learning algorithms, we first merged the full dataset and processed the EMI signals using the Root Mean Square Deviation (RMSD) index, a widely adopted metric in the field[13]. RMSD quantifies changes in the EMI signal by comparing each measurement against its baseline and has been widely used to track structural property variations, especially in concrete strength development[47–50].

Figure 3a shows the relationship between the RMSD index, calculated from the EMI signals, and the measured concrete cylinder compressive strength. When the test results from all sensors at different curing ages were combined, the data distribution lacked meaningful structure. However, when analyzing the data from a single piezoelectric sensor, we were able to replicate results similar to those reported in other studies. Specifically, the RMSD index, as it changes over time, shows a perfectly matched trend with the development of concrete strength. The reduced clarity in the data distribution when aggregating results from multiple sensors can be attributed to several factors. First, variations in sensor fabrication and piezoelectric material's intrinsic difference introduce slight differences in baseline signals. Additionally, once embedded in the concrete, factors such as sensor polarization direction, relative position within the mass concrete, and gravitational orientation all impact the signal response. Furthermore, environmental factors such as temperature changes and early hydration effects can cause fluctuations in sensor readings at

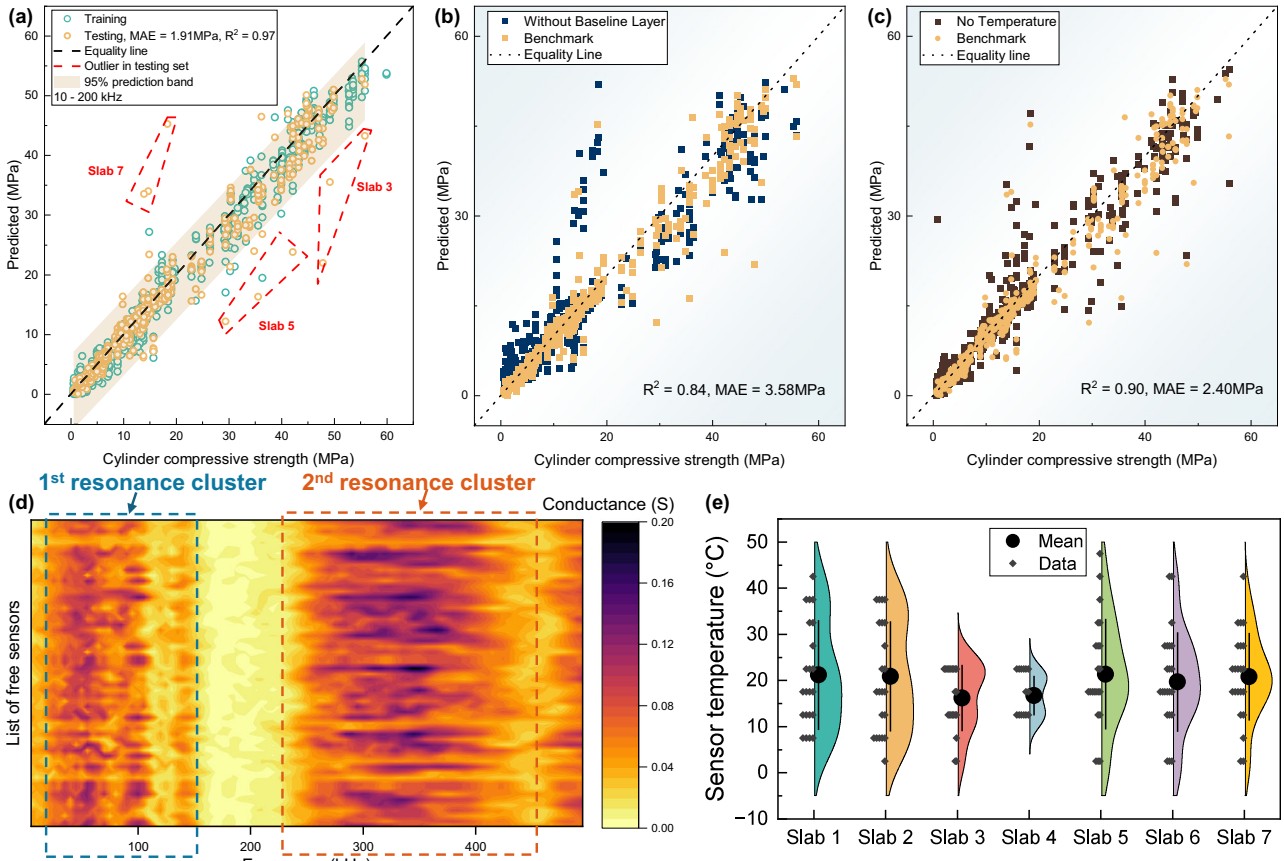

**Fig. 4 | Performance comparison of the AI model leveraging predictions from all sensors with various input features and frequency ranges. a** Benchmark model performance with the optimized frequency range (10–200 kHz). **b** Performance without the baseline mechanism. **c** Comparison of model performance with and without temperature as an input feature. **d** Contour plot of the frequency spectra for free sensors, showing two main peak clusters (10–200 kHz and 200–500 kHz) and variations between individual sensor signals, underscoring the importance of the baseline mechanism. **e** Violin plots showing sensor temperature data during slab measurements.

different time points. These challenges indicate that simple signal change quantification methods are insufficient for extracting accurate strength information from EMI signals.

In response to these issues, we designed a deep learning model using a 1DCNN architecture to address the effects of sensor discrepancies, temperature fluctuations, and other variations. The model's architecture includes a 1DCNN layer that processes both baseline and real-time signals simultaneously to capture the initial characteristics of the EMI signals. Subsequent pooling and additional 1D CNN layers preserve frequency-domain information and signal sequence. Finally, fully connected layers incorporate temperature data, allowing the model to account for its effects and predict concrete strength more effectively. We initially fed all available information into the designed model, as shown by the performance in Fig. 3(b). The dataset, which includes 7 different concrete types, 11 critical curing ages, and over 100 distinct piezoelectric sensors, contributes to the model's ability to accurately predict concrete strength, demonstrating its robustness across diverse conditions. In the testing set, the correlation between predicted and actual strength is 0.90, with a mean absolute error (MAE) of 2.69 MPa. Outliers are also highlighted, primarily from sensors in Slab 3, Slab 5, and Slab 7. These outliers likely arise because the sensors were positioned close to the CIP molds. During the year-long measurement experiment, removing the CIP molds may have influenced the signal for those sensors, meaning that the baseline signal no longer fully reflected the sensor's unaltered initial state. However, such an issue would not occur in actual service conditions, as the concrete structures in practice would not undergo intentional disturbance.

After observing the promising performance of the initial model, we proceeded with optimization and fine-tuning, resulting in the benchmark performance shown in Fig. 4. The outliers observed in Fig. 4a, consistent with those in Fig. 3b, further support the hypothesis that the removal of CIP samples influenced the sensors' EMI signals. Additionally, we tested the impact of removing key components of the model to assess their contribution. These experiments, which are summarized in Table S2 of the Supplementary information, included variations such as using different frequency ranges, and removing temperature or the baseline layer. The corresponding performance comparisons, with the optimal parameters for each configuration, are shown in Fig. 4b, c. We selected three frequency ranges for comparison: 10–200 kHz, 200–500 kHz, and 10–500 kHz. This choice was based on our observation that the resonance peaks of all signals fall within two distinct regions, shown in Fig. 4d: the first resonance cluster is in the 10–200 kHz range, and the second is in the 200–500 kHz range. This figure further supports the notion that each sensor has different initial signals, but by using the proposed model architecture, this variation can be mitigated. As detailed in Supplementary information Table S2, the model trained using signal only from the first resonance cluster consistently outperforms other frequency ranges in terms of model performance. By applying the optimal frequency range and incorporating the baseline and temperature features, we compared the proposed model with several commonly used machine learning algorithms, shown in Supplementary Fig. S20. The results showed that the proposed model achieved the best performance, with an R² of 0.94 and an MAE of 1.91. Figure 4b emphasizes the impact of

the baseline mechanism on model prediction accuracy. Without the baseline, the model's MAE nearly doubles, highlighting how the absence of a baseline leads to accumulated errors due to sensor variability, insertion location, and concrete-specific variations (such as different mix designs, varying curing conditions, and distinct concrete types), which are then reflected in the model's predictions. Figure 4c compares the importance of temperature as an input feature. While the model without temperature input still demonstrates reasonable reliability, particularly for low-strength concrete predictions, larger errors occur between 10–20 MPa. This is because concrete at this strength is typically in the 10-h to 1-day curing range, during which hydration heat accumulation causes a significant temperature to rise[51,52]. As a result, the EMI signal is influenced by both strength and temperature variations, leading to errors when temperature is not considered. Figure 4e shows the distribution of sensor temperatures across different slabs, highlighting the widespread variation due to differing concrete types and curing conditions. High-strenggth concrete predictions also show noticeable errors, likely due to temperature fluctuations during the later stages of testing (28, 90, and 180 days), as these measurements were taken during the fall season when ambient temperatures varied significantly.

## From model interpretability to understanding EMI sensing mechanisms

To interpret the underlying mechanisms of the model's concrete strength predictions, we employed Sobol sensitivity analysis. The model's inputs were categorized into four types: **C-input** (real-time concrete sensing EMI signals) mapped to strength data at the same time; **B-input** (baseline EMI signals) collected at the 4-h mark after sensor deployment to establish a baseline that compensates for potential variability from sensor fabrication, piezoelectric materials, sensor positioning, boundary conditions, and other sources; and two non-signal inputs: **Age-input** (the concrete's curing age at the time of data collection) and **T-input** (the temperature at the time of signal collection). Sobol sensitivity analysis calculates the contribution of these different input types to the strength prediction, splitting the sensitivity index into first-order contributions (the direct effect of an input on the output) and second-order contributions (interactions between inputs affecting the output). As shown in Fig. 5a, we first examined the contributions of the real-time C-input and baseline B-input across the frequency spectrum. The model identified that only frequencies below 200 kHz contribute meaningfully to strength prediction, while the higher frequency ranges had negligible impact. This frequency range aligns with the first full peak of the piezoelectric material's frequency response, even though a second peak occurs between 300–500 kHz, which the model disregards (see Fig. 4d and Supplementary Figs. S5–S18). This observation correlates with prior studies on EMI signals and concrete's elastic modulus[53], which suggest that piezoelectric sensors, when bonded with concrete, alter the electrical signal at the first resonance peak, thus reflecting changes in concrete strength. This also explains why training with data from 10–200 kHz yields the best results, as other frequency ranges provide minimal additional information for strength estimation. The B-input contributions, by contrast, are concentrated below the first major peak of the free sensor, indicating that the baseline's role is to provide a stable reference point. This data-driven analysis represents the first time that EMI signal mechanisms for structural health monitoring have been revealed, further validating prior research that links signal changes directly to structural mechanical performance. In Fig. 5b, we see that the overall contribution of the C-input is the largest, with the remaining input types (Age, Temperature, and Baseline) contributing relatively equally. This shows that the model uses auxiliary information (Age, Temperature, Baseline) in conjunction with real-time sensor data to track strength changes and make accurate strength predictions. This finding is consistent with the original model design, which aims to

eliminate the influence of disruptive information using non-real-time signal inputs. Together with the model, these features allow for accurate concrete strength extraction. Figure 5c compares the first-order and second-order contributions, which are approximately equal. This suggests that the model incorporates interactions between different inputs during prediction, leveraging combined information to produce its output. To further analyze these second-order interactions, Fig. 5d, e visualizes the contributions of different input interactions using heatmaps. The heatmaps are divided into four sections: C-input to C-input interactions, B-input to B-input interactions, C-input to B-input interactions, and interactions between non-signal inputs (Age, Temperature) and signal inputs (C and B). First, it is clear that C-C input and C-B input interactions account for the majority of contributions. C-C input interactions demonstrate how the convolutional layers capture the signal's primary features, especially those within the 200 kHz range, reflecting the characteristics of the piezoelectric sensor's first major peak and enabling strength prediction. C-B input interactions, on the other hand, showcase the importance of the baseline mechanism we introduced. The model actively combines real-time signals with baseline signals to eliminate disruptive variation sources, thereby ensuring accurate strength prediction. B-B input interactions, however, are almost negligible, as expected, since the baseline primarily functions as a compensatory mechanism rather than directly influencing prediction. The model's ability to integrate the baseline mechanism with convolutional layers demonstrates how global features are extracted for accurate predictions. Moreover, T-input and Age-input show significant interactions with C-input in the heatmaps, demonstrating that non-signal inputs are used to account for the effects of hydration or environmental temperature variations. This allows for temperature compensation, ensuring accurate strength predictions despite temperature fluctuations during concrete curing.

## Field deployment and performance evaluation

To conduct a more robust cross-validation, we collected signals under real-world construction conditions and used the model trained on all slab test data to assess concrete strength, thereby evaluating the model's performance in practical applications. As shown in Fig. 6a, we conducted four tests across two construction sites near Indianapolis, replicating the piezoelectric sensor-based concrete strength sensing experiment. All data used for strength prediction and concrete strength measurements were collected using the same methodology across the tests. These two construction sites represent common pavement concrete tasks: paving at I-74 and patching at I-465. Due to the functional differences between these tasks, the concrete used in each application had different performance characteristics. As shown in Fig. 6b, the early-age temperature of the patching concrete at I-465 is higher than that of the paving concrete at I-74. This is because patching concrete is used for road repairs and requires rapid strength gain to minimize traffic delays and congestion, leading to higher hydration heat compared to paving concrete. For this reason, I-74 allowed for strength data collection at both 1 day and 3 days post-pour, while at I-465, the test had to be completed within a single day due to the quick re-opening of the construction site, and therefore no 3-day data was available.

The model, trained on all slab test data, was then used to assess concrete strength, evaluating its performance in real-world construction project applications. In this external validation, the field data served exclusively as the test set, and the corresponding sensor profiles were entirely excluded from the training process. This strict separation ensured that the model had no prior exposure to the characteristics of these sensors or their associated concrete strength developments. As a result, any predictive performance observed in this setting reflects the model's true generalization capability under unseen and realistic construction conditions, free from information leakage. The deployment method of the piezoelectric sensors, as

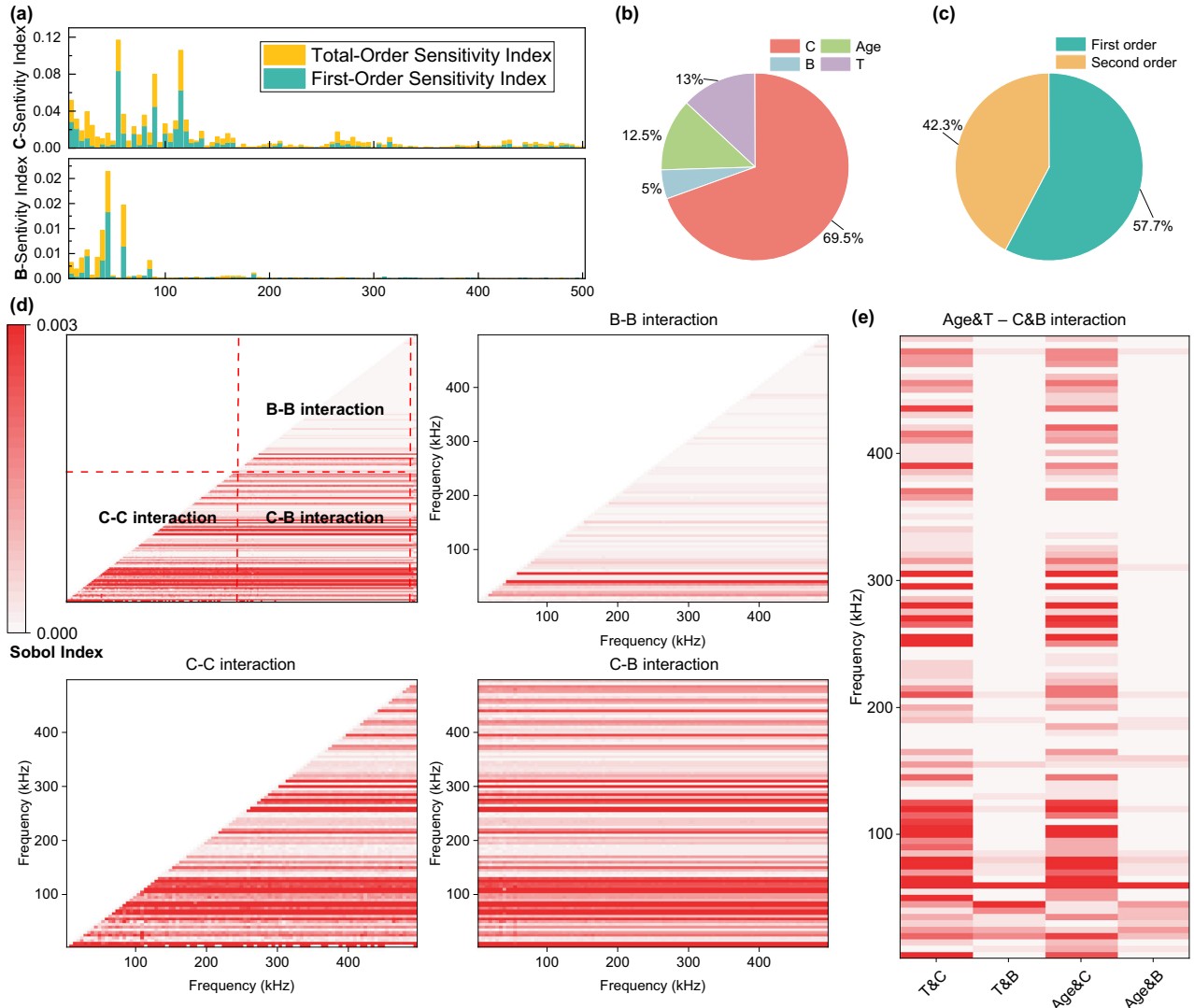

**Fig. 5 | Sensitivity analysis of the deep learning model's input contributions to concrete strength prediction. a** Sensitivity indices (first-order and total-order) for the C-input (Concrete real-time EMI signals) and B-input (baseline EMI signals), showing their contribution across the frequency spectrum. **b** Pie charts showing the overall contribution of different input types to strength prediction. C-input has the largest contribution, while B-input, Age-input (curing age of concrete), and T-input (temperature) contribute similarly. **c** Comparison of first-order and second-order sensitivity contributions. **d** Heatmaps visualizing second-order interactions between different signal input. **e** Heatmaps visualizing second-order interactions between signal input and non-signal input.

shown in Fig. 6c, involved securing the sensors onto the steel reinforcement within the concrete pavement to ensure stability. For detailed information on the placement technique, data collection process, and sampling methods, please refer to our previous work[18]. Each test involved more than six sensors, all of which provided stable signal readings. This demonstrates the feasibility of the developed technology in real construction environments, where the sensors survived the harsh conditions and did not require specialized expertise for installation. The number of sensors used per test and the average error are detailed in Supplementary Table S3. The tests showed differences in compressive strength from cylinder testing generally within ±15%, with average differences under 2.5 MPa, meeting the accuracy requirements for engineering applications.

The primary source of error likely stems from differences in the model's signal mapping. The EMI signal from sensors embedded in the patching or paving mass concrete is intended to map with the CIP mold data. However, due to limitations during the highway testing, CIP molds were not available for comparison. As a result, the model predictions were compared to cylinder test results, which typically have lower strength due to their smaller size and reduced heat retention.

This discrepancy introduces error, as shown in the time series of strength growth predictions in Fig. 6d–g, where the model predictions consistently exceed the cylinder test results. This occurs because the model is trained to predict CIP and mass concrete strength, which naturally leads to higher in-situ strength compared to cylinder samples. The actual pavement concrete benefits from better insulation and hydration heat retention, leading to higher strength, further validating the accuracy of the proposed method in reflecting in-situ concrete strength more reliably than the commonly used proxy methods. Additionally, the negligible error bands show that even with more than six sensors per prediction, the predictions remain highly consistent. The detailed results for model predictions, along with the corresponding standard deviations, can be found in Supplementary Table S4. This consistency indicates that the proposed signal processing method can account for variations in sensor placement and positioning while maintaining accurate strength predictions. Moreover, we again validated the importance of the baseline mechanism and temperature correction through external validation, as shown in Fig. S21. Compared to the benchmark model setup, predictions without these components exhibit significant errors.

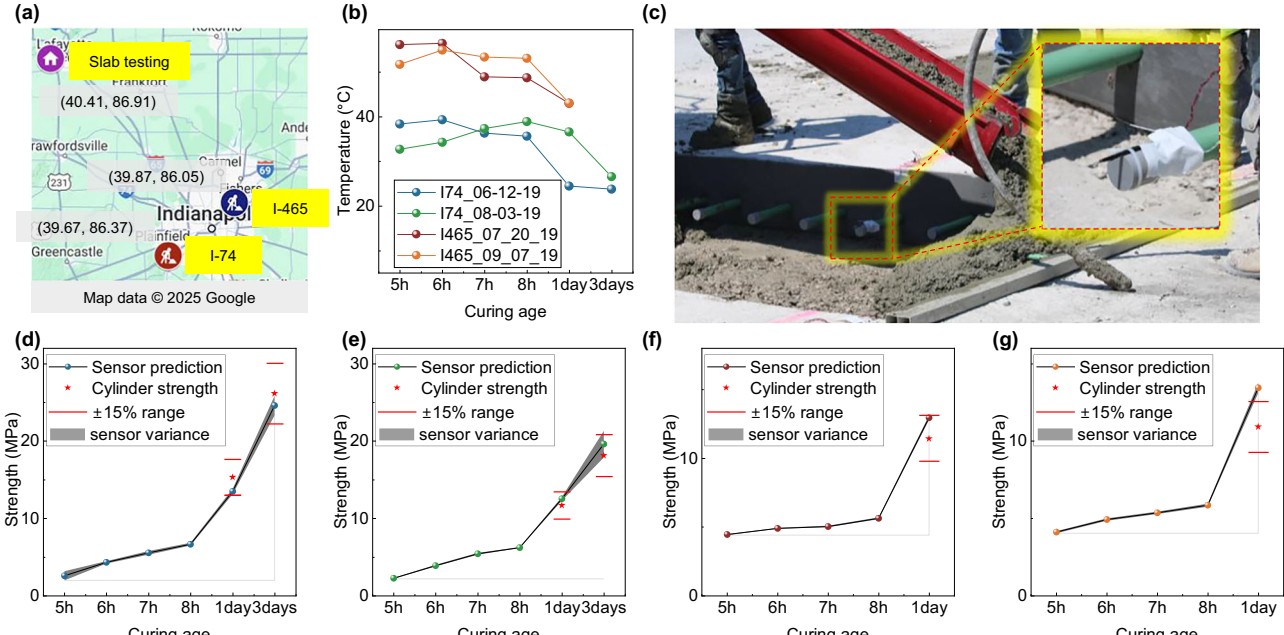

**Fig. 6 | Field test validation of the AI-assisted piezoelectric sensor signal processing method. a** Locations of the field tests on I-74 and I-465 near Indianapolis. **b** Temperature evolution during early-age curing at both sites. **c** Piezoelectric sensor deployment method for in-situ strength sensing. **d**–**g** The model predictions are compared with cylinder testing results (red stars), showing that the model tends to predict higher strength due to its mapping to mass concrete strength. Each point represents the average prediction across multiple sensors for the same slab and cylinder strength. The shaded areas represent the variance among different sensors' predictions for the same slab and cylinder strength, which are calculated based on the standard deviation of the sensor predictions, demonstrating the consistency of the model's strength predictions across multiple sensors. The ±15% range shown represents the industry acceptance margin for conventional cylinder compressive strength tests.

## Discussion

This work presents an end-to-end process, spanning from experimental research to practical deployment in real-world construction projects. We demonstrated the process of implementing a real-time concrete monitoring system by building a comprehensive database, developing the AI-assisted signal processing foundation model, and applying the model in practical scenarios. Throughout the study, we explored various model optimizations and post-processing techniques, explaining how the deep learning model interprets the mechanical properties of the host structure through piezoelectric signals. The successful implementation of this technology has led to its technical transfer and adoption as part of the new AASHTO standard (AASHTO T 412) for concrete structure strength monitoring.

Piezoelectric sensors have been investigated for structural health monitoring via electromechanical coupling since the late 1990s, but the successful translation into practical applications has not been established. Previous studies have demonstrated a consistent trend between the changes in the host structure's mechanical properties and the corresponding variations in the sensor signals. However, these efforts have often been hindered by sensor discrepancies, the lack of direct signal interpretation algorithms, and the variations introduced by diverse sensor deployments and environmental conditions, which have limited their practical application. To address these challenges, we designed an integrated approach that integrates a piezoelectric sensor with an AI-driven signal processing method. Firstly, we conducted massive testing, building a comprehensive database that maps the data from hundreds of sensors to the concrete strength profiles of various concrete types and curing conditions. This approach has resulted in a diverse and highly adaptable database capable of accommodating different construction scenarios, paving the way for more reliable and scalable concrete strength monitoring systems.

Leveraging this comprehensive database, we designed a 1DCNN-based deep learning network to develop advanced signal processing methods through a data-driven approach. The 1DCNN architecture effectively handles frequency-domain information, allowing the model to learn relevant patterns in the EMI signals. Additionally, we introduced a baseline layer design, which accommodates variations from different sensor profiles, device placements, and other complex sources of variation. Furthermore, we incorporated temperature as an additional input, enabling the deep learning model to learn how to perform temperature correction and decouple temperature-induced signal changes, leading to accurate compressive strength sensing. The results confirm that the proposed model architecture outperforms other configurations, demonstrating its superior performance.

On a mechanistic level, our work is the first to use a data-driven approach to elucidate the signal processing mechanisms of deep learning models. Through sensitivity analysis, we showed how the model structure interacts with features, incorporating the baseline to generate the final prediction. This analysis also revealed the critical role of the first resonance peak cluster in the piezoelectric sensor EMI signal, highlighting its importance in capturing meaningful structural information. These findings suggest that the model's foundation, based on data-driven insights, can be applied more broadly to structural health monitoring and non-destructive testing (NDT) in engineering applications. More importantly, the model's versatility allows it to be extended to various engineering fields, enhancing the precision and efficiency of the monitoring system for a wide range of materials and structural types.

## Methods

### Concrete slab preparation and measurement

The experiment was conducted over a one-year period, starting on 10/14/2020 (slab 1, 2), 10/20/2020 (slab 3, 4), and 11/06/2020 (slab 5, 6, 7), with measurements taken until 11/06/2021. Seven 8-foot by 12-foot concrete slabs were cast at Purdue University's Center for Aging Infrastructure (CAI), with each slab having a unique mix design. The concrete used in this experiment was supplied by IMI Irving Materials Inc., a local ready-mix company, and was based on practical concrete

pavement mix designs from the Indiana Department of Transportation (INDOT). The mix consisted of type I ordinary Portland cement (OPC), INDOT #23 fine aggregate (natural sand passing through a 3/8" sieve), and INDOT #8 coarse aggregate (gravel AP, passing a 1" sieve). A water reducer was also used, and the water-to-cement ratio varied between 0.38 and 0.45. To compare the compressive strength of the concrete, two types of samples were prepared: cast-in-place (CIP) cylinders (4" × 6") following ASTM C873, and field-molded cylinders (4" × 8") following ASTM C39[9,40]. During the tests, curing conditions such as ambient temperature, and concrete internal temperature were recorded.

## Data collection

Each concrete slab was embedded with a total of 15 piezoelectric sensors. Of these, 12 sensors were pre-positioned within the slab molds before the concrete was poured, while the remaining 3 sensors were embedded into the field-cured cylinder samples. The sensor data collected corresponded to the compressive strength data from the CIP and 4 × 8 cylinder samples, ensuring that the sensors experienced identical curing conditions as the mechanical test samples. The specific layout and placement of the sensors can be seen in Fig. 1(b), ensuring consistent data collection across the slabs and cylinder samples.

The EMI signals and concrete strength were measured at key time intervals: 6–24 h, and on days 3, 7, 28, 90, 180, and 365. The strength data and EMI signals were collected simultaneously, with a maximum time difference of 30 min to ensure accurate mapping between sensor data and mechanical performance. Since EMI signal collection before the 24-h mark was partially automated, the relatively slower mechanical strength measurements were interpolated to align with the EMI data. The EMI measurements were performed using an Arduino-integrated multiplexer and EMI measurement device, controlled by custom software. Internal temperature was continuously recorded using a multichannel data logger connected to embedded thermocouples. The frequency range of the EMI signal collection spanned from 10 kHz to 500 kHz, with a 5 kHz step size for high-resolution data acquisition.

## Database establishment

The database used in this study consists of 1,733 data points, representing the EMI signals collected from various piezoelectric sensors at critical curing ages of the concrete. Each data point comprises the following features:

- Baseline Input Features: EMI signals collected at the 4-h curing age, used for calibration against environmental variables and sensor-specific differences during the construction process.
- Real-time Concrete Sensing Input Features: EMI signals collected at the current time, reflecting changes in the signal due to the development of concrete strength.
- Temperature Feature: The internal temperature of the concrete at the time of signal collection.
- Age: The current curing age of the concrete.
- Label: The compressive strength of the concrete.

During model training and internal validation, the data was split into a training set (70%) and a test set (30%). The same random state was used for each comparison to ensure consistency in the data split, allowing for reproducibility of the results.

## Model training

The model architecture used in this study is based on a 1DCNN implemented using the PyTorch framework, with certain modifications tailored to the specific requirements of the experiment. The network consists of two convolutional layers designed to process both baseline and real-time EMI signal features. The outputs of the convolutional layers are combined with non-signal features (such as temperature and curing age) in a Multi-Layer Perceptron (MLP) layer to make the final prediction. The detailed architecture is outlined in Table 1, and the source code is available on GitHub. For each training session, the reported results were optimized through an extensive grid search to determine the best hyperparameters. The hyperparameter search and the ranges explored for the learning rate, optimizer, batch size, and layer configurations are detailed in Table 2. The model training was performed using standard backpropagation, with the best-performing set of parameters selected for final evaluation.

## Model evaluation and analysis

In this study, two primary evaluation metrics were employed to assess the performance of the model: the coefficient of determination (R²) and the mean absolute error (MAE). These metrics allow for a comprehensive evaluation of the model's ability to predict concrete strength. The **R²** score measures the correlation between the predicted and actual values in the test dataset. This metric evaluates how well the model is able to describe the trend of strength development in the concrete over time. It is defined as Eq. 1:

$$R^2 = 1 - \frac{\sum (y_i - \hat{y}_i)^2}{\sum (y_i - \bar{y}_i)^2} \qquad (1)$$

where $y_i$ are the actual values; $\hat{y}_i$ are the predicted values; $\bar{y}_i$ is the mean of the actual values. On the other hand, the MAE provides a direct and interpretable measure of how much the predicted concrete strength deviates from the actual strength, as determined through destructive testing. It is calculated as Eq. 2:

$$MAE = \frac{1}{n} \sum |y_i - \bar{y}_i| \qquad (2)$$

where n is the total number of data points.

In the field test evaluation, we used error bands based on the standard deviation to assess the consistency of strength predictions from multiple sensors deployed on the same concrete pavement section. This method evaluates the variability of sensor predictions, providing insights into the spread of results across multiple measurements. The standard deviation ($\sigma$) is calculated using the Eq. 3:

$$\sigma \sqrt{\frac{1}{n} \sum_{i=1}^{n} (s_i - \mu)^2} \qquad (3)$$

where $s_i$ represents the prediction from the i-th sensor; $\mu$ is the mean of all sensor predictions; n is the total number of the sensor predictions.

To assess the significance and contribution of each input feature, a variance-decomposition-based sensitivity analysis known as Sobol' sensitivity analysis (SSA) was applied[54]. In this approach, the model is treated as a black-box, and only the input-output relationships are considered for the analysis. SSA provides both first-order sensitivity indices and total-order sensitivity indices. The first-order indices represent the contribution of each input variable acting independently, while the total-order indices reflect the combined contribution of a variable and its interactions with other inputs. The model $Y = f(x_1, \ldots, x_m)$, where $f$ is a trained deep learning model, and $x_i$ being the input features, was analyzed using SSA. The trained function can then be decomposed as Eq. 4:

$$f(x) = f_0 + \sum_{i=1}^{N} f_i(x_i) + \sum_{i=1}^{N} \sum_{i \neq j}^{N} f_{ij}(x_i, x_j) + f_{1\ldots N}(x_1, \ldots, x_M) \qquad (4)$$

## Table 1 | 1DCNN model components

| Model Component | Description | Parameters |
|---|---|---|
| Input Layer | 1D convolutional layer that processes the input EMI signal, extracting local features. | Input channels: Inputs, Kernel size: 2 |
| Max Pooling Layer | Down samples the feature map from the input layer, reducing its spatial dimension. | Pooling size: 1 |
| Conv Layer 1 | Second convolutional layer that applies a new set of filters to further extract relevant features. | Kernel size: 1 |
| Conv Layer 2 | Third convolutional layer to refine feature extraction from the EMI signals. | Kernel size: 1 |
| Flatten Layer | Flattens the multi-dimensional feature map from the convolutional layers into a 1D vector. | |
| Linear Layer 1 | First fully connected layer reducing the dimension of the feature map. | |
| Linear Layer 2 | Second fully connected layer that further reduces the feature vector dimension. | |
| Concatenation Layer | Combines non-EMI features (e.g., temperature, curing age) with the learned features. | Concatenates features with 2 non-signal features (temperature, age) |
| Linear Layer 3 | Combines the concatenated features and reduces the final feature vector dimension. | Input size: 34, Output size: 16 |
| Output Layer | Produces the final output predicting the concrete strength. | Input size: 16, Output size: Outputs |
| Activation Function | ReLU activation is applied after each layer to introduce non-linearity. | Applied after each convolutional and fully connected layer |
| Additional Features | Includes non-EMI features such as temperature and curing age. | Non-EMI inputs: Temperature and Curing Age |
| Total Trainable Parameters | Number of trainable weights in the model. | Example: ~200k parameters (adjustable) |

## Table 2 | Hyperparameter grid search

| Hyperparameter | Values Searched |
|---|---|
| Learning Rate | 0.00001, 0.00005, 0.0001, 0.001, 0.01 |
| Optimizer | Adam, SGD |
| Batch Size | 16, 32, 64, 128 |
| Conv1 Out Channels | 16, 32, 64, 128 |
| Conv2 Out Channels | 16, 32, 64, 128 |
| Linear1 Out Features | 16, 32, 64, 128 |
| Linear2 Out Features | 16, 32, 64, 128 |

where $f_0$ is the mean value of $f(x)$ as shown in Eq. 5, and the expression of $f_i(x_i)$ and $f_{ij}(x_i, x_j)$ are listed as, Eqs. 6, 7:

$$f_0 = \int_0^1 f(x) \mathrm{d}x \tag{5}$$

$$f_i(x_i) = \int_0^1 f(x) \prod_{M \neq j} \mathrm{d}x_M - f_0 \tag{6}$$

$$f_{ij}(x_i, x_j) = \int_0^1 f(x) \prod_{k \neq i,j} \mathrm{d}x_M - f_0 - f_i(x_i) - f_j(x_j) \tag{7}$$

Squaring both sides and integrating results in the decomposition of variance as below:

$$V = \int_0^2 f^2(x)\mathrm{d}x - f_0^2 = \sum_{i=1}^M V_i + \sum_{i<j} V_{ij} + \cdots + V_{1,2,\ldots,M} \tag{8}$$

where $V(Y)$ is the total variance of the model output, and $V_{ij}$ represents the partial variance due to the interaction of $x_i$ and $x_j$. The Sobol' sensitivity indices are then defined as

$$S_i = \frac{V_i}{V(Y)} \tag{9}$$

$$S_{ij} = \frac{V_{ij}}{V(Y)} \tag{10}$$

These indices quantify the contribution of each subset of inputs to the model's output variance. For example, $S_i$ represents the first-order contribution of the i-th input; $S_{ij}$ represents the second-order interaction between the i-th and j-th inputs.

### Field test validation

In this study, four field tests were conducted as part of ongoing projects on U.S. highways I-74 and I-465, focusing on collecting identical data under real-world conditions. These tests were specifically chosen to assess the performance of the developed AI-driven piezoelectric sensor system for concrete strength monitoring in two types of highway construction environments. I-74 was used for pavement construction, while I-465 focused on patching operations. The consistency of the data collection methods across the different sites ensures the reliability and transferability of the developed system for varied construction types.

### Data availability

All data generated and analyzed in this study are openly available. The source data used for visualization are provided within the paper and its Supplementary information. The datasets used for model training and post-processing have been deposited in our GitHub repository: https://github.com/hguangshuai/EMI-Net/tree/main/Data.     Source data are provided with this paper.

### Code availability

The code used for data processing, feature extraction, and model development in this study has been deposited in Zenodo under the: https://doi.org/10.5281/zenodo.17604613.

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

## Acknowledgements
This work was supported by a Small Business Innovation Research (SBIR) Phase I grant from the National Science Foundation (NSF). Additional support was provided in part by the Joint Transportation Research Program (JTRP 4210), administered by the Indiana Department of Transportation and Purdue University, United States, as well as by the Federal Highway Administration (FHWA) through the Transportation Pooled Fund Program TPF-5(281). Dr. Han acknowledges Ms. Shiyue Wang for her assistance with figure preparation.

## Author contributions

G.H. wrote the original manuscript. Y.S. designed the experiments. G.H.Y.S., C.H., R.H., and Z.K. participated in the experiments, data collection, and analysis. G.H. was responsible for code implementation and model development. G.H. and Y.S. performed data processing. G.L., Y.F., and N.L. supervised the project. N.L., as the PI of the project, conceptualized the research, secured the funding and resources, and provided intellectual and supervisory guidance through the entire project. All authors participated in the analysis and discussion of the results.

## Competing interests
The authors declare no competing interests.
