## [Transparent Peer Review file · Nature Communications]

Real-Time Concrete Strength Monitoring Using Piezoelectric Sensors and Deep Learning

Corresponding Author: Professor Na Lu

Version 0:

Reviewer comments:

Reviewer #1

(Remarks to the Author)

This study presents a comprehensive methodology for estimating concrete compressive strength using a 1D-CNN model with piezoelectric sensor. The testing methodology, data collection, and analysis are well-structured and clearly articulated. The authors effectively outline the current limitations of non-destructive methods for evaluating concrete strength in field conditions, highlighting various challenges and sources of uncertainty. Notably, this study addresses those uncertainties and adopts an appropriate validation approach with relevant references to support the proposed data collection method.

Additionally, the use of Sobol index analysis to examine parametric contributions (e.g., baseline, temperature, and real-time signal) offers valuable insights into the contribution of each parameter. Although the field validation faces practical limitations, the authors provide reasonable justifications, which strengthens the credibility of the findings.

Overall, this paper is well-written and makes a meaningful scientific contribution. I recommend acceptance with minor revision.

Below are a few comments and questions for clarification:

Comments and Questions

Q1.

In Table S1, could you include the target compressive strength for each mix design case?

This addition would help readers understand the variability in test scenarios and the deviations between mix design expectations and actual results from casting and testing. It would also add an interesting dimension to the discussion in Section 2(d).

Q2.

Could you clarify what RMSD refers to? (Please define the acronym and provide)

Q3.

Please provide more details on the instrumentation setup, including the specific devices used, the type and specifications of the piezoelectric sensors, and the sampling frequency employed in this study.

Q4.

Could you elaborate on the PCA analysis? For instance, was the input signal raw data, or was it pre-processed in some way?

Q5.

The use of a kernel size of 1 in the convolutional layers raises questions, as it does not capture temporal or spatial correlations. It appears that only the first layer is responsible for extracting local patterns across input channels. Could the authors clarify the rationale behind this design choice and how the model parameters were determined?

(Remarks on code availability)

Reviewer #2

(Remarks to the Author)

The authors report on a sensing methodology based on electro-mechanical impedance (EMI) for monitoring concrete during curing. The ultimate goal is to estimate concrete strength based on the EMI measurements. The authors claim that they have produced a methodology that is capable of predicting strength in real-time by collecting not only EMI from the concrete, but also a reference EMI measurement and temperature, and predict strength based on a neural network regression analysis from a set of laboratory slabs. The work is interesting and important. I do believe this paper needs to be significantly revised before it can be considered for publication. I base this recommendation on the following observations"

- The paper in its current form reads like a chronological review of the work being performed over the last seven years. I don't believe this is appropriate, as it is repetitive (e.g. all plots except one in Figure 6 were shown previously) and some observations are presented and then corrected later (e.g., discussion about frequency response shown in Figure 5c, compared to Discussion section). There are also a number of unnecessary and broad summaries at the end of sections that praise the technology and what it can do (bottom of Page 5, Page 6, bottom of Page 16, top half of Page 20, top half of Page 23). Another example is the several attempts to demonstrate the accuracy of the prediction sprinkled throughout the paper. Why is the field validation not presented at the end, to make a final case for your system? The section "Model benchmarking analysis" is something that should come early on, not at the end. Please reorganize your paper in a scientific manner and avoid any non-technical summaries until the end of the paper.

- Stating that you "sense strength" is dishonest. One cannot directly do that, even using cores does not give you "actual or true strength". Strength is always implied, in this case from a dynamic measurement. So in reality you "predict strength using EMI measurements". This should be fixed in all plots where the x-axis says "actual". This was done via cylinder test, correct? Then that's what it is: Cylinder compressive strength.

- You state the limitations of other methods (Bottom of Page 3) that not all concretes are covered, but technically applies to your method as well, correct? Your method is not completely generalizable across all possible concrete mixes! Please explain.

- There is no sketch of the actual sensor, nor is the sensing principle described adequately. What does it look like? What is the setup? Please include a short description.

- You claim that your sensor (sensing method) has been adapted by AASHTO T-412. I believe that it shows the fundamental equation that relates f_c' to the static modulus of elasticity. In your work here you use a neural network, which, unless I am missing something, is not based on any equations (although a physics based neural network could be used) and is thus not the same as what is proposed in the standard. Please explain!

- The strength predictions should be given in terms of a mean with prediction limits (e.g., at the 95% level). Right now in your field validation you only show a mean prediction. You state that you get consistent predictions for multiple sensors, which means you have high precision. What you are not showing is accuracy. This could be done by using a standard procedure given in, e.g., <https://scholar.afit.edu/cgi/viewcontent.cgi?article=1180&context=facpub>. For example, adding 95% prediction limits to your plots (e.g., Figure 3b, Figure 6a, b, c) would show what kind of accuracy your predictions have. These limits could then be added in your field prediction plots. This would be honest and probably explain the discrepancy you find in your field measurements. Please consider implementing this or some other form of error/uncertainty measure to your predictions!

- Your discussion about mechanistic insights is still fairly symptomatic, and mostly discussing how and not why the signals change over time. Please elaborate!

- Make sure all figures have proper axis labeling (e.g., Figure 6d)!

(Remarks on code availability)

Only the code but no data are available.

Reviewer #3

(Remarks to the Author)

The manuscript presents a significant advancement in non-destructive testing (NDT) of concrete using piezoelectric sensors and deep learning. While the topic addressed in this manuscript is of significant interest to the civil engineering and infrastructure monitoring communities, and the integration of AI with EMI-based piezoelectric sensing is timely and potentially impactful, I find the current version of the manuscript difficult to follow due to major issues with clarity, organization, and methodological transparency. The manuscript suffers from considerable problems in writing style. Many sentences are ambiguous or vague, and key concepts are either poorly explained or repeated inconsistently. There are multiple instances where it is unclear which specific aspect of the study is being described, whether referring to model inputs, sensor configurations, data collection points, or performance metrics. Furthermore, overly rhetorical language (e.g.,

“groundbreaking,” “paradigm shift”) is used liberally without sufficient scientific support, which detracts from the technical rigor of the work. However, the most critical point is the methodology, which is not clearly explained. This makes it difficult to follow the interpretation of the results. As a result, the paper cannot be accepted in its current form. The authors are encouraged to substantially improve the methodological section of the work.

(Remarks on code availability)

Reviewer #4

(Remarks to the Author)

This study introduces an EMI-based monitoring framework integrated with machine learning algorithms for assessing the compressive strength of concrete. The manuscript proposes a one-dimensional Convolutional Neural Network designed to predict the compressive strength of concrete by utilizing the EMI responses of piezoelectric transducers, along with temperature measurements, as inputs. The authors employed both experimental data and real-world application datasets to assess the efficacy of the proposed framework, yielding promising outcomes. The research has been thoroughly designed and executed. The study's findings are encouraging and worthy of publication. However, the manuscript requires revisions before it can be deemed suitable for publication.

a) There are some typos in the manuscript, like on page 2, "However, Most NDT techniques demand specialized expertise". Please, revise the manuscript thoroughly.

b) The adopted structure of the manuscript is somewhat unconventional and confusing. The authors are encouraged to consider revising the manuscript to enhance its readability and the overall impact of the study.

c) The implementation of machine learning techniques for EMI-based Structural Health Monitoring received minimal attention in the Introduction section. The authors are advised to include a paragraph that discusses previously published studies that have employed machine learning techniques, such as hierarchical clustering and convolutional neural networks (CNN), for damage identification and the assessment of strength gain in concrete through PZT EMI responses. Furthermore, large-scale tests have already been documented in the existing literature, whereas field trials are relatively uncommon. In light of this, kindly contemplate revising the statement within the manuscript.

d) In the Materials section, the authors state, "To compare the compressive strength of the concrete, three types of samples were prepared: cast-in-place (CIP) cylinders (4" × 6") following ASTM C873, and field-molded cylinders (4" × 8") following ASTM C39". The third type appears to be missing. Kindly verify and make the necessary corrections.

e) In the Data collection section, the authors refer to an Arduino-integrated multiplexer and EMI measurement device. Please provide a more detailed explanation. Kindly specify the manufacturer or model of the EMI measurement device, the type of multiplexer employed, and how its usage may have influenced the accuracy of the EMI measurements.

f) The 70-30 ratio for data splitting is a widely adopted validation protocol in the published literature. However, due to the nature of the underlying problem, the use of a stricter validation protocol, such as the leave-one-specimen-out cross-validation, which has been used for 1-D CNN for damage classification of fiber-reinforced concrete (FRC), will enhance the applicability of the proposed approach.

(Remarks on code availability)

Version 1:

Reviewer comments:

Reviewer #1

(Remarks to the Author)

The authors have satisfactorily addressed all review comments from the initial review. Specifically:

1. Test setup and results: The authors provided detailed information including mix design specifications and strength test results. This additional insight helps readers understand practical variability in concrete mix design and field casting conditions, generating a valuable dataset that reflects real-world conditions rather than overly controlled laboratory conditions.

2. Instrumentation documentation: Additional figures and schematics of the data acquisition system and measurement setup were provided, improving the reproducibility of the experimental methodology.

3. Analysis methodology: The authors provided detailed information about their PCA analysis and CNN model design, including the rationale for parameter selections with proper comparisons and explanations of their decision-making process. The revised manuscript is suitable for publication and I recommend acceptance.

(Remarks on code availability)

Reviewer #2

(Remarks to the Author)

The authors have made a significant effort in addressing my comments, which I appreciate. The flow of the paper is much better, and inflated language has been mostly removed. At the same time, some problems have remained unaddressed.

Major comment: I do still think how the results and predictions are reported should be improved. I asked about adding 95% prediction limits to Fig. 3b that could be used to provide a measure of uncertainty to predict compressive strength. You decided not to do that (see Fig. 4a), but instead add "15% lines" to Figs. 4b and c. What is their basis? The lines originate at [0, 0], then spread out. I don't think these lines are appropriate as they clearly do not seem to capture the variance in the dataset, particularly at low concrete strengths. Finally, I don't understand how the grey shaded bands in Figs. 6d through g can be so narrow, given the variance in the dataset observable in Fig. 4a (this is the final model, correct?). Please explain!

A few additional comments:

- At the top of Page 2 you still claim that "This technology has been formally adopted by the American Association of State Highway and Transportation Officials (AASHTO) as a new national standard (AASHTO T412) marking it the first AI-powered non-destructive testing (NDT) solution to achieve such regulatory integration", which is not accurate. Please revise this statement so that it is consistent with what you state at the bottom of Page 6. Please revise!

- Fig. 3 is repeated (without caption) on Pages 14, 15, and 17, which I believe was not intended. Please fix!

- On Page 23, bottom, you mention 15% in conjunction with "field tests". You also added 15% lines to your model predictions (Fig. 4b and c), which seems incorrect and is confusing. Please explain!

(Remarks on code availability)

N/A

Version 2:

Reviewer comments:

Reviewer #2

(Remarks to the Author)

Please see the attached document.

(Remarks on code availability)

Response to Referees Letter (Han et al 2025, Nature Communications)

We sincerely thank the editors and all four reviewers for their constructive and thoughtful feedback on our manuscript entitled “*Sensing the Unsensed: AI Interprets Piezoelectric Whispers from Concrete*” (NCOMMS-25-23198). We are grateful for the reviewers’ recognition of the importance and potential impact of our work, and we appreciate their efforts in identifying key areas for improvement.

We carefully reviewed all the comments and made extensive revisions in response. Among the many valuable suggestions, two major issues were raised by multiple reviewers: (1) the need for a clearer and more organized manuscript structure, and (2) insufficient discussion in the methodology section. We fully agree with these assessments and have substantially revised the manuscript to address them. Specific responses and corresponding changes are provided in the following point-by-point section.

To aid the review process, reviewer comments are *italic*, and our responses follow each in regular text. In the revised manuscript, the changes corresponding to each comment are highlighted in blue, and whenever applicable, we quote the updated text directly. We hope the revisions address all concerns and improve both the clarity and rigor of the work. We thank the reviewers again for their valuable feedback and the opportunity to improve the manuscript.

Reviewer #1

Reviewer #1 summary:

This study presents a comprehensive methodology for estimating concrete compressive strength using a 1D-CNN model with piezoelectric sensor. The testing methodology, data collection, and analysis are well-structured and clearly articulated. The authors effectively outline the current limitations of non-destructive methods for evaluating concrete strength in field conditions, highlighting various challenges and sources of uncertainty. Notably, this study addresses those uncertainties and adopts an appropriate validation approach with relevant references to support the proposed data collection method.

Additionally, the use of Sobol index analysis to examine parametric contributions (e.g., baseline, temperature, and real-time signal) offers valuable insights into the contribution of each parameter. Although the field validation faces practical limitations, the authors provide reasonable justifications, which strengthens the credibility of the findings.

Overall, this paper is well-written and makes a meaningful scientific contribution. I recommend acceptance with minor revision.

Below are a few comments and questions for clarification:

Response: We sincerely thank the reviewer for the encouraging comments and constructive suggestions. Our detailed point-by-point responses are provided below.

Reviewer #1 comment 1:

In Table S1, could you include the target compressive strength for each mix design case?

This addition would help readers understand the variability in test scenarios and the deviations between mix design expectations and actual results from casting and testing. It would also add an interesting dimension to the discussion in Section 2(d).

Response: We thank the reviewer for this insightful suggestion and fully agree. As suggested, we have added both the target 28-day compressive strength and the measured 28-day compressive strength for each mix design in the updated Table S1. These targets were based on the Indiana Department of Transportation (INDOT) pavement design specifications and were purposefully adjusted within a controlled range to construct a diverse and practically meaningful dataset.

Table S1 Detailed mix design (lbs/cubic yards) parameters for each of the seven concrete slabs, including water-to-cement (W/C) ratios, types and quantities of admixtures, and supplementary cementitious materials (SCMs). Each slab features a distinct combination of materials to capture a wide range of strength development behaviors. All designs were based on a target 28-day compressive strength of 4,000 psi (approximately 27.58 MPa).

Slab No.	Cement	#8 Gravel AP	Natural Sand	Water reducer(oz)	Water(gal)	Nano-silica(oz)	Fly ash	28 days compressive strength (MPa)
1	563	1927	1506	38.6	16.6	22.6	0	43.21
2	563	1927	1506	38.6	18.6	22.6	0	38.48
3	564	1893	1513	22.6	21.2	22.6	0	47.26
4	450	1927	1506	22.6	18.6	0	113	\
5	568	1920	1506	11.3	16.8	0	0	33.23
6	582	1880	1493	22.3	18.3	0	0	38.60
7	558	1893	1493	22.6	22.0	0	0	13.90

To further clarify this variability, we have also added the following discussion in the revised manuscript: “These mix designs were based on standard pavement concrete mixtures commonly used by the Indiana Department of Transportation (INDOT), with a target 28 days-compressive strength of 4,000 psi (approximately 27.58 MPa). Slabs 5 exhibited compressive strengths closest to the target, as their compositions most closely followed typical INDOT specifications. Slabs 1–3 showed significantly higher 28-day strengths due to the incorporation of nano-silica, a material previously shown in our studies to enhance strength development³⁹. Slab 7, on the other hand, was deliberately designed with a higher water content to simulate underperforming or failed concrete conditions in practical construction scenarios.

Within this practical design framework, we made controlled adjustments to the proportions of key materials to generate slabs with distinct strength development profiles. This strategy enabled us to test the generalizability of the proposed method across different concrete with various mixtures. Environmental conditions for each casting date, including ambient temperature and wind speed, are shown in Fig. S2 of the Supplementary information. Each slab was cast with a unique mix design on one of three separate dates, under varying environmental conditions. By incorporating realistic variation in both curing conditions and mix composition, we aimed to closely replicate field casting scenarios. This approach allowed us to build a representative and variable-rich dataset that reflects the diverse factors influencing concrete strength in real-world construction environments.”

These additions help illustrate the relationship between mix design intent and realized performance, enhancing the context for the study's findings. We believe that the revised version now offers more detailed information and a clearer explanation of the design rationale, and we hope this adequately addresses the reviewer’s suggestion.

Reviewer #1 comment 2:

Could you clarify what RMSD refers to? (Please define the acronym and provide)

Response: We thank the reviewer for pointing out this oversight. We have now added the full-term Root Mean Square Deviation (RMSD) upon its first mention in the main text to ensure clarity for all readers: “Before applying deep learning algorithms, we first merged the full dataset and processed the EMI signals using the Root Mean Square Deviation (RMSD) index, a widely adopted metric in the field¹³”

In addition, we have included a detailed explanation of the RMSD index in the Supplementary Information, outlining its definition, how it is calculated, and the rationale behind selecting it as a comparative statistical index in our analysis:

Statistical Index for EMI Signal Analysis

To establish a baseline understanding of the EMI signal variation during strength development, we employed the Root Mean Square Deviation (RMSD) index, which quantifies the difference between a real-time EMI measurement and its baseline signal. Mathematically, RMSD is defined as:

$$RMSD\ index = \sqrt{\frac{\sum_{i=1}^N (G_r - G_{bl})^2}{\sum_{i=1}^N (G_{bl})^2}} \tag{1}$$

where G_r and G_{bl} is the conductance signal amplitude at frequency i from real-time and baseline measurements, respectively. Our previous research demonstrated that, among various quantitative indices

used in EMI-based monitoring, RMSD exhibited the highest correlation with concrete strength development, making it the most effective choice for signal characterization in this context².

Reviewer #1 comment 3:

Please provide more details on the instrumentation setup, including the specific devices used, the type and specifications of the piezoelectric sensors, and the sampling frequency employed in this study.

Response: We appreciate the reviewer’s suggestion and fully acknowledge the lack of sufficient detail in the original manuscript. To address this concern, we have added two dedicated sections in the Supplementary Information:”

S1. Sensor Design and Working Mechanism

The piezoelectric sensor used in this study is based on a PZT ceramic plate (PQYY+0412), fabricated from lead zirconate titanate (PZT) with dimensions of 10 mm × 10 mm in width and length, and 0.2 mm in thickness. The electrodes are formed by sputter-coating both sides of the PZT plate with a CuNi (Copper-Nickel) alloy, serving as the core signal transmission elements. To ensure durable electrical connection, cold soldering was performed using silver epoxy, which minimizes thermal damage to the piezoelectric material. As shown in Fig. S22, the sensing unit is then encapsulated using a polyester coating, which offers mechanical protection and insulation when embedded in concrete. This design has been validated in previous studies¹ to preserve the sensor’s electromechanical sensitivity while withstanding the harsh alkaline and high-humidity environment during concrete curing.

The integrated sensing mechanism relies on the EMI (Electro-Mechanical Impedance) principle, in which the PZT transducer serves as both an actuator and a sensor. When excited by a swept sinusoidal voltage signal, the sensor generates localized vibrations that interact with the surrounding concrete. The response is then recorded as an EMI signature, which reflects the mechanical impedance of the host structure. Changes in concrete stiffness and mass properties—associated with strength development—alter the EMI spectrum, thus enabling non-destructive monitoring of structural evolution in real time. This electromechanical interaction can be simplified into a one-dimensional model, as illustrated on the right side of Fig. S22. The underlying principle is that when the PZT sensor is bonded to a host structure, its electrical admittance response is influenced by the mechanical impedance of the surrounding material through electromechanical coupling. As concrete undergoes hydration, its mechanical impedance evolves due to changes in stiffness and mass. This variation is transferred to the PZT sensor and is reflected as a

shift or amplitude change in the EMI spectrum. Therefore, by continuously monitoring the EMI signal during early-age curing, we can effectively capture the strength development of the concrete structure.

S2. Signal Acquisition System Configuration

To support high-throughput EMI measurements across multiple embedded sensors, we developed a custom-built multiplexer (MUX) system interfaced with the AIM 4300 Impedance Analyzer (Array Solutions). As shown in the schematic and hardware photos (Fig. S and Fig. S), the MUX supports up to 16 sensor input channels and enables sequential excitation and acquisition of signals from each sensor. The switching control is managed by an Arduino Nano Every (ABX00033), programmed to autonomously coordinate timing, signal routing, and data synchronization. The relays, actuated by logic commands, are governed via an I²C-controlled GPIO expander (MCP23017) and cascaded shift registers (74HC595), allowing efficient addressable selection of each input channel. A Python-based host-side controller handles USB serial communication and timing, modeled after legacy ASCII-based protocols to ensure reliability and flexibility in field operations. To minimize signal degradation and environmental interference, the entire electronics system—including relays, Arduino controller, and supporting power circuitry—was housed in a shielded ValuLine metal chassis. This design mitigated electrical noise and provided robust protection from temperature variation, dust, and moisture during outdoor deployment.

Each measurement cycle involved applying an alternating current (AC) excitation signal of 1 V peak-to-peak to the bonded PZT patch. The impedance spectrum was scanned across a frequency range of 10 kHz to 500 kHz, with a resolution of 5 kHz, enabling high-fidelity capture of frequency-dependent EMI response. The multiplexer maintained a crosstalk level below 0.04%, and although mechanical relay switching introduced a latency of ~10–20 ms, this had negligible impact on the resolution and timing of the EMI measurements.

In addition, **three supplementary figures** have been provided to support this description:

- One showing the **sensor structure and encapsulation**,
- One showing the **overall signal acquisition architecture**, and
- One showing the **detailed MUX circuit diagram**.

Fig. S22 PZT sensor fabrication, installation, and its coupling with concrete structure.

Fig. S23 Schematic of the multi-channel signal collection signal

Fig. S24 Circuit design of the MUX

Reviewer #1 comment 4:

Could you elaborate on the PCA analysis? For instance, was the input signal raw data, or was it pre-processed in some way?

Response: We sincerely thank the reviewer for this insightful comment. In response, we have added a dedicated section in the Supplementary Information that provides a detailed explanation of the PCA analysis:

S4. Principal Component Analysis (PCA) for Database-Wide Signal–Strength Relationship

To explore the relationship between EMI signals and concrete strength across the entire database, Principal Component Analysis (PCA) was performed on the collected EMI spectra. PCA is a linear dimensionality reduction technique that transforms high-dimensional data into orthogonal principal components (PCs), ranked by the amount of variance each component captures. Each EMI signal vector, consisting of amplitudes across frequencies, was projected into a 2D space defined by the first two principal components (PC_1 and PC_2). The percentage values associated with each axis (e.g., PC_1: 65.41%) indicate the proportion of total variance in the original data explained by that component.

We compared two PCA projections: one using only the real-time EMI signals, and the other incorporating both baseline and real-time signals as inputs. The projection using only real-time signals showed scattered and irregular distributions, making it difficult to discern strength-related patterns. In contrast, the inclusion of baseline signals significantly improved the compactness and structure of the projection, with data points showing clearer trends related to strength progression. This further supports the necessity of the baseline mechanism to reduce sensor-to-sensor variation and enhance model interpretability.

Reviewer #1 comment 5:

The use of a kernel size of 1 in the convolutional layers raises questions, as it does not capture temporal or spatial correlations. It appears that only the first layer is responsible for extracting local patterns across input channels. Could the authors clarify the rationale behind this design choice and how the model parameters were determined?

Response: We sincerely thank the reviewer for this sharp and technically insightful comment. The observation is correct and aligns with a key design consideration during our model development. In fact, we explored numerous configurations in the early design phase—including whether to use 1D CNN layers at all, whether to use kernel sizes of 1, and whether to rely solely on linear layers. Each component in the final model architecture reflects a choice based on empirical optimization and domain-specific constraints.

To clarify:

- The first convolutional layer uses a kernel size of 2, which is critical for jointly processing baseline and real-time EMI signals at the input level. This design allows the model to begin integrating features from both input channels early on.
- Subsequent convolutional layers use a kernel size of 1, which may appear unconventional but serves two primary purposes:
 - (1) Unlike standard linear layers, 1D CNN layers with kernel size 1 retain the order of features (frequency bins), which is important for preserving frequency-domain patterns;
 - (2) They offer parameter efficiency, which is essential given our data size—approximately 1700 instances, which is at the lower bound of feasibility for deep learning applications. Adding larger kernels or deeper structures led to overfitting, which we empirically observed during architecture search.

To directly address the reviewer’s concern, we attached two ablation experiments and results shown below:

One removes the kernel size 1 convolutional layers (C1 and C2), replacing them with linear layers. The other removes the two fully connected layers (L1 and L2) before the addition of temperature input. In both cases, prediction performance **significantly declined** compared to the benchmark model, validating the importance of these components in the final architecture.

In the main text, we deliberately did not include detailed comparisons of alternative model structures to avoid losing focus on the core sensing methodology (baseline and additional temperature input). However, we acknowledge the reviewer’s point that more explanation on the model architecture design is helpful. While architectural optimization (e.g., kernel size choices) was not emphasized in the manuscript,

we have already provided a full summary of model parameters and optimization strategies (such as frequency range and hyperparameter tuning via grid search) in the Supplementary Information (Table S2). To address the reviewer's concern, we have also added a more explicit discussion of our architectural rationale in the revised main text.:”we designed a deep learning model using a 1DCNN architecture to address the effects of sensor discrepancies, temperature fluctuations, and other variations. The model structure is as follows: A 1DCNN layer with processes both baseline and real-time signals simultaneously to capture the initial characteristics of the EMI signals. This is followed by pooling layers and additional 1D CNN layers, which preserve frequency-domain information while maintaining the sequential order of the signal. Finally, fully connected layers incorporate temperature data, enabling the model to account for its effects and predict concrete strength more effectively”

Reviewer #2

Reviewer #2 summary:

The authors report on a sensing methodology based on electro-mechanical impedance (EMI) for monitoring concrete during curing. The ultimate goal is to estimate concrete strength based on the EMI measurements. The authors claim that they have produced a methodology that is capable of predicting strength in real-time by collecting not only EMI from the concrete, but also a reference EMI measurement and temperature, and predict strength based on a neural network regression analysis from a set of laboratory slabs. The work is interesting and important. I do believe this paper needs to be significantly revised before it can be considered for publication. I base this recommendation on the following observations

Response: We sincerely thank the reviewer for recognizing the importance and potential impact of our work. We appreciate your thoughtful and constructive comments. In response to your observations, we have carefully revised the manuscript and provide detailed point-to-point responses below to address each concern.

Reviewer #2 comment 1:

- The paper in its current form reads like a chronological review of the work being performed over the last seven years. I don't believe this is appropriate, as it is repetitive (e.g. all plots except one in Figure 6 were shown previously) and some observations are presented and then corrected later (e.g., discussion about frequency response shown in Figure 5c, compared to Discussion section). There are also a number of unnecessary and broad summaries at the end of sections that praise the technology and what it can do (bottom of Page 5, Page 6, bottom of Page 16, top half of Page 20, top half of Page 23). Another example is the several attempts to demonstrate the accuracy of the prediction sprinkled throughout the paper. Why is the field validation not presented at the end, to make a final case for your system? The section "Model benchmarking analysis" is something that should come early on, not at the end. Please reorganize your paper in a scientific manner and avoid any non-technical summaries until the end of the paper.

Response: Thank you for this valuable feedback. We fully acknowledge that the original structure of the manuscript lacked clarity and coherence in terms of scientific narrative. In the revised version, we have thoroughly reorganized the manuscript into the following sections to better reflect a logical and scientific progression: Full-Scale Experimentation and Data Infrastructure → Model Development and

Optimization → From Model Interpretability to Understanding EMI Sensing Mechanisms → Field Deployment and Performance Evaluation.

This revised structure allows us to clearly present the work from experimental data collection to model design, followed by model interpretation and practical application. We agree this reorganization improves both readability and scientific rigor. In addition, we have removed broad, non-technical summaries that previously appeared at the end of sections and ensured that field validation is now presented as the final part of the study, aligning with the reviewer’s suggestion to build a cohesive narrative that culminates in real-world validation.

Regarding the reviewer’s concern about redundancy, we agree that Figures 3 and 4(a) may appear visually similar. However, they serve distinct purposes:

- Figure 3(b) was used to show the baseline performance improvement introduced by incorporating AI over traditional statistical indicators (like RMSD), without fine-tuning or model selection.
- Figure 4(a) presents the benchmark result after optimization and is used to compare the model against alternative configurations and input feature sets.

To prevent confusion, we have added additional clarification in the revised manuscript to explain these differences more clearly:” Before applying deep learning algorithms, we first merged the full dataset and processed the EMI signals using the Root Mean Square Deviation (RMSD) index, a widely adopted metric in the field¹³. RMSD quantifies changes in the EMI signal by comparing each measurement against its baseline and has been widely used to track structural property variations, especially in concrete strength development⁴⁷⁻⁵⁰”

Reviewer #2 comment 2:

- Stating that you "sense strength" is dishonest. One cannot directly do that, even using cores does not give you "actual or true strength". Strength is always implied, in this case from a dynamic measurement. So in reality you "predict strength using EMI measurements". This should be fixed in all plots where the x-axis says "actual". This was done via cylinder test, correct? Then that's what it is: Cylinder compressive strength.

Response: We fully agree with the reviewer’s observation—strength in this context is inferred through standardized mechanical testing, not directly measured. Accordingly, we have revised all relevant figures and captions to replace the term “actual strength” with “Cylinder compressive strength”, which more accurately reflects the reference data used for model validation.

Reviewer #2 comment 3:

- You state the limitations of other methods (Bottom of Page 3) that not all concretes are covered, but technically applies to your method as well, correct? Your method is not completely generalizable across all possible concrete mixes! Please explain.

Response: Thank you for raising this important concern. We acknowledge that our original discussion in this section was not sufficiently nuanced. The comparison we made was intended to highlight the challenges faced by another increasingly popular AI-based approach—predicting concrete strength from chemical composition data. As discussed, this method is inherently difficult to generalize due to regional variations in raw materials (such as cement chemistry, aggregate type, and water quality), making reliable inference across geographies and mix designs extremely challenging. In contrast, our approach is fundamentally different: it relies on direct mechanical-to-mechanical mapping using EMI signals captured by piezoelectric sensors. This avoids the indirect inference required by chemistry-based models and instead reflects real-time mechanical behavior of the material.

Therefore, in the revised manuscript, we've made the changes as follow: “In addition to leveraging AI for processing EMI signals, research efforts have also attempted to use AI to predict concrete performance based on mix design^{35,36}. Yet, the existing mix-design based method has inherent restrictions. In principle, these approaches are built on the assumption that a given mix design will always result in similar chemical composition, hydration behavior, and, ultimately, strength development. However, this assumption does not hold in real-world scenarios, where binder type, aggregate source, water quality, and environmental exposure can significantly alter hydration kinetics—even when the nominal mix design remains the same. For instance, supplementary cementitious materials (SCMs) and admixtures used in concrete have countless variations, each with distinct chemical compositions and varying impacts on concrete performance^{23,37}. Additionally, the chemical composition of cement and the properties of fine and coarse aggregates vary greatly by region, making it difficult to generalize models suitable for different geographies³⁸. As a result, the development of universally applicable databases and predictive models built on chemical composition inputs remains highly challenging and often ineffective for field conditions. In contrast to mix-design-based approaches, piezoelectric sensing combined with electromechanical coupling enables a direct mechanical-to-mechanical mapping. Mechanistically, the EMI signal captures changes arising solely from variations in the mechanical properties of the host structure, without relying on assumptions about chemical composition or hydration behavior. As a result, the extracted information

is inherently unaffected by variations in chemical composition, curing conditions, or other human factors, making it robust, broadly applicable, and well-suited for diverse field conditions.”

We have also made extensive efforts to test the method on different types of concrete, as documented in the report available at [<https://rosap.ntl.bts.gov/view/dot/54753>] and summarized in the table below, which outlines the range of concrete types evaluated using the same sensing methodology.

mixture	mean	std	min	max
unit: lbs/cuyd				
cement	527.62	155.36	208.00	949.99
water	259.28	43.67	147.16	402.53
fly_ash	142.45	45.32	50.00	278.00
slag	179.94	51.23	108.75	312.00
silica_fume	27.00	0.00	27.00	27.00
fibers	4.50	0.00	4.50	4.50
coarse_aggregate	1816.01	181.29	1300.00	2395.49
fine_aggregate	1318.65	154.90	966.00	1811.41
water_reducer	3.79	6.45	0.14	52.87
superplasticizer	1.52	0.17	0.78	1.56
accelerator	12.27	14.93	0.52	45.64
retarder	10.52	18.85	0.19	68.85
air_entrainer	0.41	0.31	0.12	1.80
e5_internal_cure	1.88	1.31	0.25	4.68
liquid_fly_ash	2.26	1.92	0.51	4.01
quikrete	3387.89	257.38	3016.26	3824.10
rapid_cement	704.14	158.40	275.00	907.00

Reviewer #2 comment 4:

- There is no sketch of the actual sensor, nor is the sensing principle described adequately. What does it look like? What is the setup? Please include a short description.

Response: Thank you for pointing this out. In response, we have added a dedicated section in the Supplementary Information that provides a detailed description of the sensor design and working principle, along with three supporting figures illustrating the sensor design and signal collection system. The following content has been added into the revised manuscript:

S1. Sensor Design and Working Mechanism

The piezoelectric sensor used in this study is based on a PZT ceramic plate (PQYY+0412), fabricated from lead zirconate titanate (PZT) with dimensions of 10 mm × 10 mm in width and length, and 0.2 mm in thickness. The electrodes are formed by sputter-coating both sides of the PZT plate with a CuNi (Copper-Nickel) alloy, serving as the core signal transmission elements. To ensure durable electrical connection, cold soldering was performed using silver epoxy, which minimizes thermal damage to the piezoelectric material. As shown in Fig. S, the sensing unit is then encapsulated using a polyester coating, which offers mechanical protection and insulation when embedded in concrete. This design has been validated in previous studies¹ to preserve the sensor's electromechanical sensitivity while withstanding the harsh alkaline and high-humidity environment of curing concrete.

The integrated sensing mechanism relies on the EMI (Electro-Mechanical Impedance) principle, in which the PZT transducer serves as both an actuator and a sensor. When excited by a swept sinusoidal voltage signal, the sensor generates localized vibrations that interact with the surrounding concrete. The response is then recorded as an EMI signature, which reflects the mechanical impedance of the host structure. Changes in concrete stiffness and mass properties—associated with strength development—alter the EMI spectrum, thus enabling non-destructive monitoring of structural evolution in real time. This electromechanical interaction can be simplified into a one-dimensional model, as illustrated on the right side of Fig. S. The underlying principle is that when the PZT sensor is bonded to a host structure, its electrical admittance response is influenced by the mechanical impedance of the surrounding material through electromechanical coupling. As concrete undergoes hydration, its mechanical impedance evolves due to changes in stiffness and mass. This variation is transferred to the PZT sensor and is reflected as a shift or amplitude change in the EMI spectrum. Therefore, by continuously monitoring the EMI signal during early-age curing, it is possible to capture the strength development of the concrete.

S2. Signal Acquisition System Configuration

To support high-throughput EMI measurements across multiple embedded sensors, we developed a custom-built multiplexer (MUX) system interfaced with the AIM 4300 Impedance Analyzer (Array Solutions). As shown in the schematic and hardware photos (Fig. S and Fig. S), the MUX supports up to 16 sensor input channels and enables sequential excitation and acquisition of signals from each sensor. The switching control is managed by an Arduino Nano Every (ABX00033), programmed to autonomously coordinate timing, signal routing, and data synchronization. The relays, actuated by logic commands, are

governed via an I²C-controlled GPIO expander (MCP23017) and cascaded shift registers (74HC595), allowing efficient addressable selection of each input channel. A Python-based host-side controller handles USB serial communication and timing, modeled after legacy ASCII-based protocols to ensure reliability and flexibility in field operations. To minimize signal degradation and environmental interference, the entire electronics system—including relays, Arduino controller, and supporting power circuitry—was housed in a shielded ValuLine metal chassis. This design mitigated electrical noise and provided robust protection from temperature variation, dust, and moisture during outdoor deployment.

Each measurement cycle involved applying an alternating current (AC) excitation signal of 1 V peak-to-peak to the bonded PZT patch. The impedance spectrum was scanned across a frequency range of 10 kHz to 500 kHz, with a resolution of 5 kHz, enabling high-fidelity capture of frequency-dependent EMI response. The multiplexer maintained a crosstalk level below 0.04%, and although mechanical relay switching introduced a latency of ~10–20 ms, this had negligible impact on the resolution and timing of the EMI measurements.

In addition, **three supplementary figures** have been provided to support this description:

- One showing the **sensor structure and encapsulation**,
- One showing the **overall signal acquisition architecture**, and
- One showing the **detailed MUX circuit diagram**.

Fig. S22 PZT sensor fabrication, installation, and its coupling with concrete structure.

Fig. S23 Schematic of the multi-channel signal collection signal

Fig. S24 Circuit design of the MUX

Reviewer #2 comment 5:

- You claim that your sensor (sensing method) has been adapted by AASHTO T-412. I believe that it shows the fundamental equation that relates f_c' to the static modulus of elasticity. In your work here you use a neural network, which, unless I am missing something, is not based on any equations (although a physics based neural network could be used) and is thus not the same as what is proposed in the standard. Please explain!

Response: We thank the reviewer for raising this important point. Our work is an enhancement of the AASHTO T-412 method and is intended to complement rather than conflict with it. The AASHTO standard outlines a procedure for estimating the dynamic modulus of elasticity from EMI measurements, which is then used to infer compressive strength via empirical equations. In contrast, our method uses the same sensing mechanism but replaces the empirical mapping with a machine learning approach.

Fundamentally, both approaches rely on extracting information from the first resonance peak of the piezoelectric EMI spectrum to estimate strength. However, our neural network-based method can more effectively handle variations across sensors, installation conditions, and temperature fluctuations—challenges that often undermine the accuracy of empirical relationships. This makes our method an enhancement of the AASHTO framework, maintaining the same underlying while significantly improving robustness, accuracy, and scalability in real-world applications.

To clarify this distinction, we have revised the manuscript to use more precise language regarding the connection between this method and the AASHTO standard: “While the core sensing principle has informed the development of the American Association of State Highway and Transportation Officials (AASHTO, T 412-24) standard, the piezoelectric-based concrete strength sensing method presented here is currently undergoing field trials in over 34 U.S. states, offering a more efficient solution for highway pavement projects.”

Reviewer #2 comment 6:

- The strength predictions should be given in terms of a mean with prediction limits (e.g., at the 95% level). Right now, in your field validation you only show a mean prediction. You state that you get consistent predictions for multiple sensors, which means you have high precision. What you are not showing is accuracy. This could be done by using a standard procedure given in, e.g., <https://scholar.afit.edu/cgi/viewcontent.cgi?article=1180&context=facpub>. For example, adding 95% prediction limits to your plots (e.g., Figure 3b, Figure 6a, b, c) would show what kind of accuracy your predictions have. These limits could then be added in your field prediction plots. This would be honest and probably explain the discrepancy you find in your field measurements. Please consider implementing this or some other form of error/uncertainty measure to your predictions!

Response: We sincerely thank the reviewer for this constructive suggestion regarding the presentation of prediction uncertainty. We fully agree that visualizing prediction accuracy and uncertainty is critical for a fair and transparent evaluation of the model's performance, especially in field validation.

In response, we have implemented the suggested changes. Specifically, we have added $\pm 15\%$ prediction band in multiple figures. Two example figures with the updated visualization are also attached below for reference:

For the sensor consistence, as shown in the revised Figs. 4 (b-c) and Figs. 6(d–g), the shaded error bands represent the sensor-to-sensor variability, with the corresponding statistics reported in Table S4. While these deviations are relatively small (typically within ± 1 MPa), we have now clearly annotated them in the revised figures to highlight this information.

We appreciate the reviewer’s recommendation, which has helped us strengthen the presentation and further validate the robustness of our approach.

Reviewer #2 comment 7:

- *Your discussion about mechanistic insights is still fairly symptomatic and mostly discussing how and not why the signals change over time. Please elaborate!*

Response: We thank the reviewer for this insightful comment. To address this, we have significantly expanded the discussion in both the main manuscript and the Supplementary Information. The discussion related to signal change are revised to: “Fig.2 (c) presents the sensing results from a representative piezoelectric sensor. Before the sensor was deployed in the concrete structure, **two main resonance peak clusters** are observed in the 5-500 kHz frequency range. Once the sensor was embedded in the concrete structure, we began signal collection at the 4-hour mark of the concrete curing process, using this time point as the baseline for strength prediction. Prior research has demonstrated that signal collection at 4

hours can effectively track concrete strength development patterns¹³. In the baseline signal, the peaks diminish significantly, displaying very low amplitudes. This is because, at early ages, concrete behaves more like a viscoelastic material, absorbing most of the mechanical waves emitted by the piezoelectric sensor^{45,46}. As the concrete continues to cure, exceeding the 10-hour mark, the signal peaks start to change, evolving from a single peak to twin-peak or multi-peak patterns. This transition indicates that the concrete is transforming from a viscoelastic material to an elastic one, with reduced wave energy absorption^{44,45}. These results demonstrate that the piezoelectric sensor's EMI signals can effectively capture changes in the mechanical properties of the concrete. **This correlation arises from the electromechanical coupling principle underlying EMI sensing: as the mechanical impedance of the host material (i.e., the concrete) evolves during curing, it alters the boundary conditions experienced by the piezoelectric patch. These changes in mechanical impedance directly affect the electrical admittance spectrum of the sensor, especially the amplitude and position of its resonance peaks. Thus, variations in concrete stiffness and damping characteristics are encoded in the EMI signal, enabling real-time monitoring of mechanical property development.** Additional EMI signals collected over time from the seven concrete slabs and different sensors, showing how the signals evolve with age, can be found in the Supplementary information, Fig. S5 – S18.”

Reviewer #2 comment 8:

- Make sure all figures have proper axis labeling (e.g., Figure 6d)!

Response: We thank the reviewer for this comment. We have revised the figures and make sure all figures have proper axis labeling.

Reviewer #2 comment 9 (Remarks on code availability):

-Only the code but no data are available.

Response: We would like to clarify that the associated input data required for running the model are also provided in the same GitHub repository in .h5 format. Detailed instructions for using these data files are included in the repository’s documentation.

To further improve clarity, we have updated the Data Availability section in the revised manuscript:

“Data availability

The source data used for visualization in this study are provided within the paper and its supplementary materials. The data used for model training and post-processing are available at our GitHub repository:

<https://github.com/hguangshuai/EMI-Net/tree/main/Data>. All other data supporting this project are available from the corresponding author upon reasonable request.”

Reviewer #3

Reviewer #3 summary:

The manuscript presents a significant advancement in non-destructive testing (NDT) of concrete using piezoelectric sensors and deep learning. While the topic addressed in this manuscript is of significant interest to the civil engineering and infrastructure monitoring communities, and the integration of AI with EMI-based piezoelectric sensing is timely and potentially impactful, I find the current version of the manuscript difficult to follow due to major issues with clarity, organization, and methodological transparency. The manuscript suffers from considerable problems in writing style. Many sentences are ambiguous or vague, and key concepts are either poorly explained or repeated inconsistently. There are multiple instances where it is unclear which specific aspect of the study is being described whether referring to model inputs, sensor configurations, data collection points, or performance metrics. Furthermore, overly rhetorical language (e.g., “groundbreaking,” “paradigm shift”) is used liberally without sufficient scientific support, which detracts from the technical rigor of the work. However, the most critical point is the methodology, which is not clearly explained. This makes it difficult to follow the interpretation of the results. As a result, the paper cannot be accepted in its current form. The authors are encouraged to substantially improve the methodological section of the work.

Response: We sincerely thank the reviewer for the constructive and detailed comments. We fully acknowledge the concerns regarding clarity, organization, and methodological transparency, and we have made extensive revisions to address these issues.

To improve overall readability and scientific rigor, we have reorganized the manuscript’s structure. The revised version now follows a clearer and more conventional scientific narrative:

- Full-Scale Experimentation and Data Infrastructure
- Model Development and Optimization
- From Model Interpretability to Understanding EMI Sensing Mechanisms
- Field Deployment and Performance Evaluation

This new structure allows readers to better follow the logical flow of the work—from experimental data collection, model building, and mechanistic analysis to final field application.

In addition, we carefully revised the writing to avoid rhetorical or exaggerated language such as “groundbreaking” and “paradigm shift.” Instead, we now emphasize the practical and scientific contributions in a more measured and technically grounded manner. The previous discussion has been revised to: “Our findings propose a novel and scalable method to integrate intelligent sensing into civil

infrastructure system. This will enable the development of resilient and sustainable infrastructure, moving beyond traditional infrastructure monitoring.”

Regarding methodological transparency, we agree this was a key area that needed improvement. To address this, we added three supplementary figures that clearly show the sensor design, signal acquisition system, and circuit-level setup (Fig. S22-Fig. S24 from the revised supplementary information). For instance:

Fig S23 Schematic of the multi-channel signal collection signal

Furthermore, we introduced four new supplementary sections that provide detailed explanations on:

S1. Sensor Design and Working Mechanism

S2. Signal Acquisition System Configuration

S3. Statistical Index for EMI Signal Analysis

S4. Principal Component Analysis (PCA) for Database-Wide Signal–Strength Relationship

We hope these comprehensive changes substantially improve the clarity and technical rigor of the manuscript and address the reviewer’s concerns.

Reviewer #4

Reviewer #4 summary:

This study introduces an EMI-based monitoring framework integrated with machine learning algorithms for assessing the compressive strength of concrete. The manuscript proposes a one-dimensional Convolutional Neural Network designed to predict the compressive strength of concrete by utilizing the EMI responses of piezoelectric transducers, along with temperature measurements, as inputs. The authors employed both experimental data and real-world application datasets to assess the efficacy of the proposed framework, yielding promising outcomes. The research has been thoroughly designed and executed. The study's findings are encouraging and worthy of publication. However, the manuscript requires revisions before it can be deemed suitable for publication.

Response: We sincerely thank the reviewer for their encouraging comments and recognition of our work. We have revised the manuscript for improved clarity and completeness, and detailed responses are provided below.

Reviewer #4 comment 1:

a) There are some typos in the manuscript, like on page 2, "However, Most NDT techniques demand specialized expertise". Please, revise the manuscript thoroughly.

Response: Thank you for pointing this out. We have thoroughly revisited the manuscript and corrected all identified typos, including the one mentioned. A careful proofreading was conducted to ensure clarity and consistency throughout the text.

Reviewer #4 comment 2:

b) The adopted structure of the manuscript is somewhat unconventional and confusing. The authors are encouraged to consider revising the manuscript to enhance its readability and the overall impact of the study.

Response: We fully agree and have reorganized the manuscript structure to improve clarity and readability. The revised version follows a more conventional scientific flow: (1) Full-Scale Experimentation and Data Infrastructure, (2) Model Development and Optimization, (3) From Model Interpretability to Understanding EMI Sensing Mechanisms, (4) Field Deployment and Performance Evaluation. We believe this revised structure better supports the narrative and improves the overall impact of the study.

Reviewer #4 comment 3:

c) The implementation of machine learning techniques for EMI-based Structural Health Monitoring received minimal attention in the Introduction section. The authors are advised to include a paragraph that discusses previously published studies that have employed machine learning techniques, such as hierarchical clustering and convolutional neural networks (CNN), for damage identification and the assessment of strength gain in concrete through PZT EMI responses. Furthermore, large-scale tests have already been documented in the existing literature, whereas field trials are relatively uncommon. In light of this, kindly contemplate revising the statement within the manuscript.

Response: Thank you for the insightful comment. We agree that the original manuscript lacked sufficient discussion of prior work related to machine learning applications in EMI-based structural health monitoring. Additionally, some of our earlier descriptions regarding the scale and novelty of existing studies were insufficient. To address this, we have revised the Introduction and added the following paragraph: “With advancements in data science, techniques such as machine learning, deep learning, and artificial intelligence (AI) have increasingly been utilized to process complex datasets and have been explored in EMI-based approaches for piezoelectric sensor monitoring^{27–30}. These approaches have been applied not only for compressive strength estimation, but also for damage detection^{31,32}, damage classification³³, and corrosion assessment³⁴ using EMI signals, with some studies further integrating AI algorithms to enhance sensitivity for understanding micro-level structural deterioration. However, most existing studies have been exclusively conducted in controlled laboratory settings, where small-scale samples eliminate the effects of temperature fluctuations and ensure uniform curing conditions. As a result, the generated datasets often lack the real-world variables in construction projects, such as sensor-to-sensor discrepancies, ambient temperature fluctuations, and changes in material properties. Moreover, these limited lab scale database approaches are typically constrained by limited sensor setups and a narrow selection of concrete types, further restricting the generalizability of the models to actual engineering conditions¹⁸.”

In addition to leveraging AI for processing EMI signals, research efforts have also attempted to use AI to predict concrete performance based on mix design^{35,36}. Yet, the existing mix-design based method has inherent restrictions. In principle, these approaches are built on the assumption that a given mix design will always result in similar chemical composition, hydration behavior, and, ultimately, strength development. However, this assumption does not hold in real-world scenarios, where binder type, aggregate source, water quality, and environmental exposure can significantly alter hydration kinetics—

even when the nominal mix design remains the same. For instance, supplementary cementitious materials (SCMs) and admixtures used in concrete have countless variations, each with distinct chemical compositions and varying impacts on concrete performance^{23,37}. Additionally, the chemical composition of cement and the properties of fine and coarse aggregates vary greatly by region, making it difficult to generalize models suitable for different geographies³⁸. As a result, the development of universally applicable databases and predictive models built on chemical composition inputs remains highly challenging and often ineffective for field conditions. In contrast to mix-design-based approaches, piezoelectric sensing combined with electromechanical coupling enables a direct mechanical-to-mechanical mapping. Mechanistically, the EMI signal captures changes arising solely from variations in the mechanical properties of the host structure, without relying on assumptions about chemical composition or hydration behavior. As a result, the extracted information is inherently unaffected by variations in chemical composition, curing conditions, or other human factors, making it robust, broadly applicable, and well-suited for diverse field conditions.”

Reviewer #4 comment 4:

d) In the Materials section, the authors state, "To compare the compressive strength of the concrete, three types of samples were prepared: cast-in-place (CIP) cylinders (4" × 6") following ASTM C873, and field-molded cylinders (4" × 8") following ASTM C39". The third type appears to be missing. Kindly verify and make the necessary corrections.

Response: Thank you for your careful reading and kind observation. We acknowledge that this was a typographical error in the original manuscript. As the reviewer correctly pointed out, only **two types** of samples were prepared: (1) cast-in-place (CIP) cylinders (4" × 6") following ASTM C873, and (2) field-molded cylinders (4" × 8") following ASTM C39. The reference to “three types” has been corrected in the revised manuscript.

Reviewer #4 comment 5:

e) In the Data collection section, the authors refer to an Arduino-integrated multiplexer and EMI measurement device. Please provide a more detailed explanation. Kindly specify the manufacturer or model of the EMI measurement device, the type of multiplexer employed, and how its usage may have influenced the accuracy of the EMI measurements.

Response: We thank the reviewer for the insightful comment and the opportunity to clarify. The multiplexer (MUX) system referenced in the Data Collection section was custom-built in-house to

interface with the EMI measurement system. The MUX was designed to accommodate up to 16 sensor input channels and served as a switching front end to the AIM 4300 Impedance Analyzer (manufactured by Array Solutions). Three MUX was built and used. (Figure shown the 8 channel one)

The MUX was controlled by an Arduino Nano Every (Model ABX00033) microcontroller, which managed relay-based channel selection through a programmable control sequence. The Arduino was programmed to operate in autonomous model allowing synchronized switching and data collection over a USB serial connection. A Python-based control program handled interface management. The communication protocol was modeled after legacy ASCII-based control schemes (similar to early HPIB instruments such as the HP5334A), ensuring reliable integration with the measurement software.

The in-house MUX design supported bandwidths up to 1 MHz, with crosstalk measured below 0.04%, as validated through preliminary calibration tests. Although the use of mechanical relays introduced minor switching latency (~10–20 ms), this had a negligible impact on measurement accuracy due to the resolution and timescale of the impedance measurements.

The entire system, including the Arduino, relays, and supporting circuitry, was enclosed in a ValuLine Chassis Enclosure, which allowed for robust operation and protection against environmental disturbances during outdoor measurements. This enclosure helped mitigate ambient electrical noise and shielded the system from moisture, dust, and temperature variations experienced in field deployments.

For the revised manuscript, we've elaborated this information in supplementary information as below:

Fig. S23 Schematic of the multi-channel signal collection signal

Fig. S24 Circuit design of the MUX

S2. Signal Acquisition System Configuration

To support high-throughput EMI measurements across multiple embedded sensors, we developed a custom-built multiplexer (MUX) system interfaced with the AIM 4300 Impedance Analyzer (Array Solutions). As shown in the schematic and hardware photos (Fig. S and Fig. S), the MUX supports up to 16 sensor input channels and enables sequential excitation and acquisition of signals from each sensor. The switching control is managed by an Arduino Nano Every (ABX00033), programmed to autonomously coordinate timing, signal routing, and data synchronization. The relays, actuated by logic commands, are governed via an I²C-controlled GPIO expander (MCP23017) and cascaded shift registers (74HC595), allowing efficient addressable selection of each input channel. A Python-based host-side controller handles USB serial communication and timing, modeled after legacy ASCII-based protocols to ensure reliability and flexibility in field operations. To minimize signal degradation and environmental interference, the entire electronics system—including relays, Arduino controller, and supporting power circuitry—was housed in a shielded ValuLine metal chassis. This design mitigated electrical noise and provided robust protection from temperature variation, dust, and moisture during outdoor deployment.

Each measurement cycle involved applying an alternating current (AC) excitation signal of 1 V peak-to-peak to the bonded PZT patch. The impedance spectrum was scanned across a frequency range of 10 kHz to 500 kHz, with a resolution of 5 kHz, enabling high-fidelity capture of frequency-dependent EMI response. The multiplexer maintained a crosstalk level below 0.04%, and although mechanical relay switching introduced a latency of ~10–20 ms, this had negligible impact on the resolution and timing of the EMI measurements.

Reviewer #4 comment 6:

f) The 70-30 ratio for data splitting is a widely adopted validation protocol in the published literature. However, due to the nature of the underlying problem, the use of a stricter validation protocol, such as the leave-one-specimen-out cross-validation, which has been used for 1-D CNN for damage classification of fiber-reinforced concrete (FRC), will enhance the applicability of the proposed approach.

Response: Thank you for this insightful and important suggestion.

We fully agree that more stringent validation protocols are essential to properly evaluate model generalizability, particularly in complex, real-world applications. We firmly believe that external validation, where entire sensor profiles are completely excluded from the training set, provides a more rigorous and realistic assessment of the model's performance.

To this end, we conducted a separate field validation using over 30 sensors deployed in real construction projects, none of which were included in the training dataset. This ensures that no information leakage occurred and that the model was tested against entirely unseen sensor characteristics and concrete strength profiles.

To clarify this in the revised manuscript, we have added the following explanation: “The model, trained on all slab test data, was then used to assess concrete strength, evaluating its performance in real-world construction project applications. In this external validation, the field data served exclusively as the test set, and the corresponding sensor profiles were entirely excluded from the training process. This strict separation ensured that the model had no prior exposure to the characteristics of these sensors or their associated concrete strength developments. As a result, any predictive performance observed in this setting reflects the model's true generalization capability under unseen and realistic construction conditions, free from information leakage.”

Response to Referees Letter (Han et al 2025, Nature Communications)

We sincerely thank the editors and all four reviewers for their constructive and thoughtful feedback on our manuscript entitled “Sensing the Unsensed: AI Interprets Piezoelectric Whispers from Concrete” (NCOMMS-25-23198A). We are grateful for the reviewers’ recognition of the importance and potential impact of our work, and we greatly appreciate their efforts in highlighting areas for improvement.

We have carefully considered all of the comments and made extensive revisions in response. To aid the review process, reviewer comments are *italic*, and our responses follow each in regular text. In the revised manuscript, **the changes corresponding to each comment are highlighted in blue**, and whenever applicable, we quote the updated text directly. We hope the revisions address all concerns and improve both the clarity and rigor of the work. We thank the reviewers again for their valuable feedback and the opportunity to improve the manuscript.

Reviewer #1

Reviewer #1 summary:

The authors have satisfactorily addressed all review comments from the initial review. Specifically:

1. Test setup and results: The authors provided detailed information including mix design specifications and strength test results. This additional insight helps readers understand practical variability in concrete mix design and field casting conditions, generating a valuable dataset that reflects real-world conditions rather than overly controlled laboratory conditions.

2. Instrumentation documentation: Additional figures and schematics of the data acquisition system and measurement setup were provided, improving the reproducibility of the experimental methodology.

3. Analysis methodology: The authors provided detailed information about their PCA analysis and CNN model design, including the rationale for parameter selections with proper comparisons and explanations of their decision-making process.

The revised manuscript is suitable for publication and I recommend acceptance.

Response: We sincerely thank the reviewer for the positive evaluation of our work and for the thoughtful effort invested during the review process. We greatly appreciate the recognition of the improvements made in the revised manuscript.

Reviewer #2

Reviewer #2 summary:

The authors have made a significant effort in addressing my comments, which I appreciate. The flow of the paper is much better, and inflated language has been mostly removed. At the same time, some problems have remained unaddressed.

Response: We sincerely thank the reviewer for the constructive guidance and for acknowledging the improvements we have made in the revised manuscript. We truly appreciate your recognition of our efforts in improving the flow and clarity of the paper. We carefully note that some concerns remain unaddressed, and in the following, we provide point-by-point responses and corresponding revisions to address each of your comments.

Reviewer #2 comment 1:

-Major comment: I do still think how the results and predictions are reported should be improved. I asked about adding 95% prediction limits to Fig. 3b that could be used to provide a measure of uncertainty to predict compressive strength. You decided not to do that (see Fig. 4a), but instead add “15% lines” to Figs. 4b and c. What is their basis? The lines originate at [0, 0], then spread out. I don’t think these lines are appropriate as they clearly do not seem to capture the variance in the dataset, particularly at low concrete strengths. Finally, I don’t understand how the grey shaded bands in Figs. 6d through g can be so narrow, given the variance in the dataset observable in Fig. 4a (this is the final model, correct?). Please explain!

Response: We thank the reviewer for these insightful comments. We agree that the use of the $\pm 15\%$ lines in the earlier version of the manuscript was not appropriate and could be misinterpreted. We have therefore reworked the figures, captions, and discussion to address this issue.

Following the reviewer’s instruction, we have added 95% prediction bands to Fig. 3b and Fig. 4a to better present the results and to provide a statistically meaningful measure of predictive uncertainty. At the same time, we have removed the use of the $\pm 15\%$ lines in these figures to avoid confusion. We fully agree with

the reviewer that applying $\pm 15\%$ lines in this context was a misuse and could cause confusion. Because

Revised Fig. 3 b with updated prediction band.

Revised Fig. 4 a-c with updated prediction band.

Figs. 3 and 4 compile all individual sensor predictions and are primarily intended to illustrate model development, the $\pm 15\%$ lines are not meaningful in this setting. In contrast, the data in Fig. 6 represent the field application scenario, where sensor predictions are first averaged within each time segment before comparison. In this context, applying the $\pm 15\%$ margin represents the industry acceptance margin. To improve clarity in the revised manuscript, we have also provided more detailed explanations in the captions of Figs. 4, 6 as bellow:

- Fig. 4 Performance comparison of the AI model leveraging predictions from all sensors with various input features and frequency ranges. (a) Benchmark model performance with the optimized frequency range (10-200 kHz); (b) Performance without the baseline mechanism; (c) Comparison of model performance with and without temperature as an input feature; (d) Contour plot of the frequency spectra for free sensors, showing two main peak clusters (10-200 kHz and 200-500 kHz) and variations between individual sensor signals, underscoring the importance of the baseline mechanism; (e) Violin plots showing sensor temperature data during slab measurements.
- Fig. 6 Field test validation of the AI-assisted piezoelectric sensor signal processing method. (a) Locations of the field tests on I-74 and I-465 near Indianapolis. (b) Temperature evolution during early-age curing at both sites; (c) Piezoelectric sensor deployment method for in-situ strength sensing; (d–g) The model predictions are compared with cylinder testing results (red stars), showing that the model tends to predict higher strength due to its mapping to mass concrete strength. Each point represents the average prediction across multiple sensors for the same slab and cylinder strength. The shaded areas represent the variance among different sensors' predictions for the same slab and cylinder strength, which are calculated based on the standard deviation of the sensor predictions, demonstrating the consistency of the model's strength predictions across multiple sensors. The $\pm 15\%$ range shown represents the industry acceptance margin for conventional cylinder compressive strength tests.

We recognize the reviewer's concern regarding the apparent discrepancy between the broader scatter in Fig. 4a and the narrower shaded regions in Fig. 6d–g, and we agree that our previous explanation was not unclear. The wider scatter in Fig. 4a arises because predictions from different sensors at different time points are aggregated together, which produces a more dispersed distribution and can give the impression

of larger inter-sensor differences. In reality, due to the design of our baseline layer, the variance attributed to the different sensors themselves is quite small. To illustrate this more clearly, we have added Fig. S19 in the Supporting Information, where the results are separated by slab. This visualization makes individual predictions more transparent and shows that, for points with the same cylinder compressive strength, the predictions from different sensors are very close to each other. This observation is consistent with the shaded regions in Fig. 6, which represent the variance among sensor predictions for the same slab and cylinder strength.

Representative figure from supporting document, Fig. S19.

To provide further clarity, we have added the following discussion in the revised manuscript: “After observing the promising performance of the initial model, we proceeded with optimization and fine-tuning, resulting in the benchmark performance shown in Fig. 4 (a). The 95% prediction band (shaded) demonstrates that more than 90% of test points fall within the interval, indicating good consistency between predicted and measured strengths. To make it clearer how different sensors behave at the same cylinder compressive strength, Fig. S19 presents the data separated by slab. It can be observed that, for cylinders with the same compressive strength, the predictions from different sensors are very close to each other, demonstrating the consistency of the model.”

We’ve also updated the caption for Fig.6 as mentioned above. We hope that these revisions make our discussion clearer and more transparent for the readers.

Reviewer #2 comment 2:

At the top of Page 2 you still claim that “This technology has been formally adopted by the American Association of State Highway and Transportation Officials (AASHTO) as a new national standard (AASHTO T412) marking it the first AI-powered non-destructive testing (NDT) solution to achieve such regulatory integration”, which is not accurate. Please revise this statement so that it is consistent with what you state at the bottom of Page 6. Please revise!

Response: We thank the reviewer for pointing this out. We have revised the sentence on Page 2 to make it consistent with the statement on Page 6. The following content has been updated:” This technology has been formally adopted by the American Association of State Highway and Transportation Officials (AASHTO) as a new national standard (AASHTO T412) — marking an important step toward national standardization of non-destructive testing (NDT) methods.”

Reviewer #2 comment 3:

Fig. 3 is repeated (without caption) on Pages 14, 15, and 17, which I believe was not intended. Please fix!

Response: We thank the reviewer for pointing this out. The repeated appearances of Fig. 3 on Pages 14, 15, and 17 were due to an error in our Word field settings. We have corrected this issue in the revised manuscript.

Reviewer #2 comment 4:

On Page 23, bottom, you mention 15% in conjunction with “field tests”. You also added 15% lines to your model predictions (Fig. 4b and c), which seems incorrect and is confusing. Please explain!

Response: Thank you for flagging this—we fully agree that our use of the $\pm 15\%$ lines could cause confusion. As also noted in our response to Comment 1, we have thoroughly redone Figs. 4 and 6 together with their captions to enhance clarity. Further explanations are as follows:

- Fig. 4b–c (model development): In the revised manuscript we removed the $\pm 15\%$ lines from Figs. 4b and 4c and instead illustrated uncertainty using the 95% prediction bands shown in Fig. 3b and 4a.
- Fig. 6 (field comparison): Here the $\pm 15\%$ margin is kept only as an industry acceptance margin based on the typical variability of cylinder compressive strength tests. It is not a statistical prediction interval of our model. In the revised version, we have updated the Fig. 6 caption to clearly state that the shaded areas show the variance among sensors, while the $\pm 15\%$ band represents the industry acceptance margin relative to cylinder tests.
- Text on Page 23: We have revised the wording to make clear that “ $\pm 15\%$ in the field tests” refers to industry acceptance margin, not a model uncertainty band.:” The tests showed differences from cylinder compressive strength generally within \$\pm 15\%\$, with average differences under 2.5 MPa, meeting the accuracy requirements for engineering applications.”

We hope these changes address the reviewer’s concerns and enhance clarity. Thanks for the opportunity to improve the quality of our manuscript.

Response to Referees Letter (Han et al 2025, Nature Communications)

We sincerely thank the editors and Reviewer #2 for the additional feedback on our manuscript entitled “Sensing the Unsensed: AI Interprets Piezoelectric Whispers from Concrete” (NCOMMS-25-23198B). We have carefully addressed all remaining comments and made the corresponding revisions. Reviewer comments are shown in italics, followed by our responses in regular text. We hope these updates fully resolve the outstanding concerns.

Reviewer #2

Reviewer #2 summary:

Specifically Thank you for further improving your article! The following comments/questions still remain open on my part::

1. The sentence “This technology has been formally adopted by the American Association of State Highway and Transportation Officials (AASHTO) as a new national standard (AASHTO T412)—marking an important step toward national standardization of non-destructive testing (NDT) methods.” is still not true. The standard does not cover the entire technology described in this paper, only parts of it. While the basic relationships are described in the standard, i.e., the assumed relationship between the dynamic modulus of elasticity and compressive strength, your AI-driven analysis is not.

Response: We sincerely thank the reviewer for pointing out this issue. We agree that the previous wording was misleading, as the AASHTO T412 standard does not encompass the entire AI-driven technology described in this paper. In response, we have revised the sentence to more accurately reflect the scope of the standard. The revised text now reads: “**The underlying sensing principle of this technology has been incorporated into a new standard by the American Association of State Highway and Transportation Officials (AASHTO T412), representing a significant step toward the national standardization of this non-destructive testing (NDT) method.**” This correction has been implemented in the revised version of the manuscript.

2. I still don’t understand how your gray prediction variances in Fig. 6d to e can be so much narrower compared to your 95% prediction limits shown in Fig. 4a. See next page for illustration and example.

Response: We thank the reviewer for this insightful comment and fully understand the concern. The apparent difference between the gray prediction variances in Fig. 6d–e and the 95% prediction limits in Fig. 4a arises because these two figures present results from different experimental conditions.

Specifically, Fig. 4 shows the combined predictions of multiple sensors for different concrete samples with varying strengths, and therefore the 95% prediction limits represent the overall model uncertainty across a wide range of compressive strengths.

In contrast, Fig. 6 presents the averaged predictions of multiple sensors installed on the same concrete sample with a uniform strength. The gray bands in Fig. 6 represent the variance among sensor predictions for that single sample, rather than the full statistical prediction interval of the regression model. Hence, the narrower range reflects the high consistency among sensors rather than model uncertainty.

To clarify this distinction, we have emphasized in both the Fig. 4 and Fig. 6 captions that Fig. 4 shows combined model predictions across all datasets, while Fig. 6 presents the averaged predictions for a uniform-strength concrete sample.

For further comparison, the results in Supplementary Fig. S19 also show that the inter-sensor variance for uniform-strength samples is very close to that observed in Fig. 6, supporting this interpretation.

We hope this explanation addresses the reviewer's concern.

3. Fig.4 still appears multiple times, making it difficult to read the paper. The revised manuscript is suitable for publication and I recommend acceptance.

Response: We sincerely thank the reviewer for the positive recommendation and for pointing out this formatting issue. We apologize for the repeated appearance of Fig. 4, which was caused by a Word cross-referencing error. This has now been corrected in the revised manuscript, and proper cross-references have been implemented to ensure that Fig. 4 appears only once.

Thank you for further improving your article! The following comments/questions still remain open on my part:

- The sentence “This technology has been formally adopted by the American Association of State Highway and Transportation Officials (AASHTO) as a new national standard (AASHTO T412)—marking an important step toward national standardization of non-destructive testing (NDT) methods.” is still not true. The standard does not cover the entire technology described in this paper, only parts of it. While the basic relationships are described in the standard, i.e., the assumed relationship between the dynamic modulus of elasticity and compressive strength, your AI-driven analysis is not.
- I still don’t understand how your gray prediction variances in Fig. 6d to e can be so much narrower compared to your 95% prediction limits shown in Fig. 4a. See next page for illustration and example.
- Fig.4 still appears multiple times, making it difficult to read the paper.

Assuming that Fig. 4a (shown above) is the model used for your predictions, and using the methodology provided in “investr: An R Package for Inverse Estimation” (which I recommended in a previous review) here’s an example of how these data would be interpreted: For a predicted strength of 20 MPa, you would get an estimate of cylinder compressive strength ranging from ~ 14 to 28 MPa (at the 95% confidence level). Why are the gray bands shown in, e.g., Fig. 6e so much narrower? Shouldn’t they be the same? How is “sensor prediction variance” calculated?